# Rank-N-Contrast: Learning Continuous Representations for Regression

Kaiwen Zha[1,*]    Peng Cao[1,*]    Jeany Son[2]    Yuzhe Yang[1]    Dina Katabi[1]

[1]MIT CSAIL    [2]GIST

## Abstract

Deep regression models typically learn in an end-to-end fashion without explicitly emphasizing a *regression-aware representation*. Consequently, the learned representations exhibit fragmentation and fail to capture the *continuous* nature of sample orders, inducing suboptimal results across a wide range of regression tasks. To fill the gap, we propose Rank-N-Contrast (RNC), a framework that learns continuous representations for regression by *contrasting* samples against each other based on their *rankings* in the target space. We demonstrate, theoretically and empirically, that RNC guarantees the desired order of learned representations in accordance with the target orders, enjoying not only better performance but also significantly improved robustness, efficiency, and generalization. Extensive experiments using five real-world regression datasets that span computer vision, human-computer interaction, and healthcare verify that RNC achieves state-of-the-art performance, highlighting its intriguing properties including better data efficiency, robustness to spurious targets and data corruptions, and generalization to distribution shifts. Code is available at: https://github.com/kaiwenzha/Rank-N-Contrast.

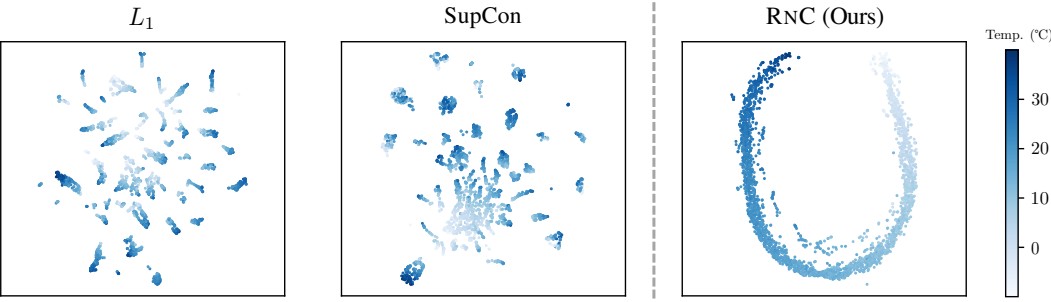

Figure 1: **Learned representations of different methods on a real-world temperature regression task** [7] (details in Sec. 5). Existing general regression learning ($L_1$) or representation learning (SupCon) schemes fail to recognize the underlying continuous information in data. In contrast, RNC learns *continuous* representations that capture the intrinsic sample orders w.r.t. the regression targets.

## 1  Introduction

Regression problems are pervasive and fundamental in the real world, spanning various tasks and domains including estimating age from human appearance [36], predicting health scores via human physiological signals [11], and detecting gaze directions using webcam images [52]. Due to the

---

*The two authors contributed equally and order was determined by a random coin flip. Correspondence to Kaiwen Zha <kzha@mit.edu>, Peng Cao <pengcao@mit.edu>.

37th Conference on Neural Information Processing Systems (NeurIPS 2023).

continuity in regression targets, the most widely adopted approach for training regression models is to directly predict the target value and employ a distance-based loss function, such as $L_1$ or $L_2$ distance, between the prediction and the ground-truth target [51, 52, 38]. Methods that tackle regression tasks using classification models trained with cross-entropy loss have also been studied [36, 33, 40].

However, previous methods focus on imposing constraints on the *final* predictions in an end-to-end fashion, but do not explicitly emphasize the ***representations*** learned by the model. Unfortunately, these representations are often fragmented and incapable of capturing the *continuous* relationships that underlie regression tasks. Fig. 1(a) highlights the representations learned by the $L_1$ loss on `SkyFinder` [7], a regression dataset for predicting weather temperature from webcam outdoor images captured at different locations (details in Sec. 5). Rather than exhibiting the continuous ground-truth temperatures, the learned representations are grouped by *different webcams* in a fragmented manner. Such unordered and fragmented representation is suboptimal for the regression task and can even hamper performance by including irrelevant information, such as the capturing webcam.

Furthermore, despite the great success of representation learning schemes on solving *discrete* classification or segmentation tasks (e.g., contrastive learning [4, 20] and supervised contrastive learning (SupCon) [25]), less attention has been paid to designing algorithms that capture the intrinsic *continuity* in data for regression. Interestingly, we highlight that existing representation learning methods inevitably overlook the continuous nature in data: Fig. 1(b) shows the representation learned by SupCon on the `SkyFinder` dataset, where it again fails to capture the underlying continuous order between the samples, resulting in a suboptimal representation for regression tasks.

To fill the gap, we present Rank-N-Contrast (RNC), a novel framework for generic regression learning. RNC first learns a *regression-aware* representation that orders the distances in the embedding space based on the target values, and then leverages it to predict the continuous targets. To achieve this, we propose the Rank-N-Contrast loss ($\mathcal{L}_{\mathrm{RNC}}$), which **ranks** the samples in a batch according to their labels and then **contrasts** them against each other based on their relative rankings. Theoretically, we prove that optimizing $\mathcal{L}_{\mathrm{RNC}}$ results in features that are ordered according to the continuous labels, leading to improved performance in downstream regression tasks. As confirmed in Fig. 1(c), RNC learns continuous representations that capture the intrinsic ordered relationships between samples. Notably, our framework is orthogonal to existing regression methods, allowing for the use of any regression method to map the learned representation to the final prediction values.

To support practical evaluations, we benchmark RNC against state-of-the-art (SOTA) regression and representation learning schemes on five real-world regression datasets that span computer vision, human-computer interaction, and healthcare. Rigorous experiments verify the superior performance, robustness, and efficiency of RNC on learning continuous targets. Our contributions are as follows:

- We identify the limitation of current regression and representation learning methods for continuous targets, and uncover intrinsic properties of learning regression-aware representations.
- We design RNC, a simple & effective method that learns continuous representations for regression.
- We conduct extensive experiments on five diverse regression datasets in *vision*, *human-computer interaction*, and *healthcare*, verifying the superior performance of RNC against SOTA schemes.
- Further analyses reveal intriguing properties of RNC on its data efficiency, robustness to spurious targets & data corruptions, and better generalization to unseen targets.

## 2   Related Work

**Regression Learning.** Deep learning has achieved great success in addressing regression tasks [36, 51, 52, 38, 11]. In regression learning, the final predictions of the model are typically trained in an end-to-end manner to be close to the targets. Standard regression losses include the $L_1$ loss, the mean squared error (MSE) loss, and the Huber loss [22]. Past work has also proposed several variants. One branch of work [36, 13, 14, 35] divides the regression range into small bins, converting the problem into a classification task. Another line of work [12, 2, 40] casts regression as an ordinal classification problem [33] using ordered thresholds and employing multiple binary classifiers. Recently, a line of work proposes to regularize the embedding space for regression, ranging from modeling feature space uncertainty [28], encouraging higher-entropy feature spaces [50], to regularizing features for imbalanced regression [44, 17]. In contrast to existing works, we provide a regression-aware representation learning approach that emphasizes the continuity in the features space w.r.t. the targets, which enjoys better performance while being compatible to prior regression schemes. In addition,

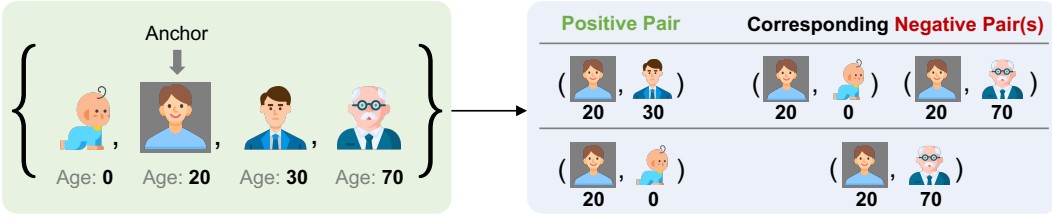

**(a)** A Batch of Samples          **(b)** Pair Construction for RNC

Figure 2: **Illustration of $\mathcal{L}_{\mathbf{RNC}}$ in the context of positive and negative pairs. (a)** An example batch of input data and their labels. **(b)** Two example positive pairs and corresponding negative pair(s) when the anchor is the 20-year-old man (shown in gray shading). When the anchor forms a positive pair with a 30-year-old man, their label distance is 10, hence the corresponding negative samples are the 0-year-old baby and the 70-year-old man, whose label distances to the anchor are larger than 10. When the 0-year-old baby creates a positive pair with the anchor, only the 70-year-old man has a larger label distance to the anchor, thus serving as a negative sample.

C-Mixup [47] adapts the original mixup [49] by adjusting the sampling probability of the mixed pairs according to the target similarities. It is also worth noting that our method is orthogonal and complementary to data augmentation algorithms for regression learning, such as C-Mixup [47].

**Representation Learning.** Representation learning is crucial in machine learning, often studied in the context of classification. Recently, contrastive learning has emerged as a popular technique for self-supervised representation learning [4, 20, 5, 8, 3]. The supervised version of contrastive learning, SupCon [25], has been shown to outperform the conventional cross-entropy loss on multiple discrete classification tasks, including image recognition [25], noisy labels [26], long-tailed classification [23, 27], and out-of-domain detection [48]. A few recent papers propose to adapt SupCon to tackle ordered labels in specific downstream applications, including gaze estimation [42], medical imaging [9, 10], and neural behavior analysis [37]. Different from prior works that directly adapt SupCon, we incorporate the intrinsic property of label continuity for designing representation learning scheme tailored for regression, which offers a simple and principled approach for generic regression tasks.

## 3 Our Approach: Rank-N-Contrast (RNC)

**Problem Setup.** Given a regression task, we aim to train a neural network model composed of a feature encoder $f(\cdot) : X \to \mathbb{R}^{d_e}$ and a predictor $g(\cdot) : \mathbb{R}^{d_e} \to \mathbb{R}^{d_t}$ to predict the target $\boldsymbol{y} \in \mathbb{R}^{d_t}$ based on the input data $\boldsymbol{x} \in X$.

For a positive integer $I$, let $[I]$ denote the set $\{1, 2, \cdots, I\}$. Given a randomly sampled batch of $N$ input and label pairs $\{(\boldsymbol{x}_n, \boldsymbol{y}_n)\}_{n \in [N]}$, we apply standard data augmentations to obtain a two-view batch $\{(\tilde{\boldsymbol{x}}_\ell, \tilde{\boldsymbol{y}}_\ell)\}_{\ell \in [2N]}$, where $\tilde{\boldsymbol{x}}_{2n} = t(\boldsymbol{x}_n)$ and $\tilde{\boldsymbol{x}}_{2n-1} = t'(\boldsymbol{x}_n)$, with $t$ and $t'$ being independently sampled augmentation operations, and $\tilde{\boldsymbol{y}}_{2n} = \tilde{\boldsymbol{y}}_{2n-1} = \boldsymbol{y}_n, \forall n \in [N]$. The augmented batch is then fed into the encoder $f$ to obtain the feature embedding for each augmented input data, i.e., $\boldsymbol{v}_l = f(\tilde{\boldsymbol{x}}_l) \in \mathbb{R}^{d_e}, \forall l \in [2N]$. The representation learning phase is then performed over the feature embeddings. To harness the acquired representation for regression, we freeze the encoder $f(\cdot)$ and train the predictor $g(\cdot)$ on top of it using a regression loss (e.g., $L_1$ loss).

In this context, a natural question arises: *How to design a regression-aware representation learning scheme tailored for continuous and ordered samples?*

**The Rank-N-Contrast Loss.** In order to align distances in the embedding space ordered by distances in their labels, we propose Rank-N-Contrast loss ($\mathcal{L}_{\mathrm{RNC}}$), which first **ranks** the samples according to their target distances, and then **contrasts** them against each other based on their relative rankings.

Following [16, 45], given an anchor $\boldsymbol{v}_i$, we model the likelihood of any other $\boldsymbol{v}_j$ to increase exponentially with respect to their similarity in the representation space [15]. Inspired by the listwise ranking methods [43, 6], we introduce $\mathcal{S}_{i,j} := \{\boldsymbol{v}_k \mid k \neq i, d(\tilde{\boldsymbol{y}}_i, \tilde{\boldsymbol{y}}_k) \geq d(\tilde{\boldsymbol{y}}_i, \tilde{\boldsymbol{y}}_j)\}$ to denote the set of samples that are of higher **ranks** than $\boldsymbol{v}_j$ in terms of label distance w.r.t. $\boldsymbol{v}_i$, where $d(\cdot, \cdot)$ is the distance measure between two labels (e.g., $L_1$ distance). Then the normalized likelihood of $\boldsymbol{v}_j$ given $\boldsymbol{v}_i$ and $\mathcal{S}_{i,j}$ can be written as

$$\mathbb{P}(\boldsymbol{v}_j | \boldsymbol{v}_i, \mathcal{S}_{i,j}) = \frac{\exp(\mathrm{sim}(\boldsymbol{v}_i, \boldsymbol{v}_j)/\tau)}{\sum_{\boldsymbol{v}_k \in \mathcal{S}_{i,j}} \exp(\mathrm{sim}(\boldsymbol{v}_i, \boldsymbol{v}_k)/\tau)}, \tag{1}$$

where $\text{sim}(\cdot, \cdot)$ is the similarity measure between two feature embeddings (e.g., negative $L_2$ norm) and $\tau$ denotes the temperature parameter. Note that the denominator is a sum over the set of samples that possess *higher ranks* than $\boldsymbol{v}_j$; Maximizing $\mathbb{P}(\boldsymbol{v}_j | \boldsymbol{v}_i, \mathcal{S}_{i,j})$ effectively increases the probability that $\boldsymbol{v}_j$ outperforms the other samples in the set and emerges at the top rank within $\mathcal{S}_{i,j}$.

As a result, we define the per-sample RNC loss as the average negative log-likelihood over all other samples in a given batch:

$$l_{\text{RNC}}^{(i)} = \frac{1}{2N-1} \sum_{j=1,\ j\neq i}^{2N} -\log \frac{\exp(\text{sim}(\boldsymbol{v}_i, \boldsymbol{v}_j)/\tau)}{\sum_{\boldsymbol{v}_k \in \mathcal{S}_{i,j}} \exp(\text{sim}(\boldsymbol{v}_i, \boldsymbol{v}_k)/\tau)}. \tag{2}$$

Intuitively, for an anchor sample $i$, *any* other sample $j$ in the batch is **contrasted** with it, enforcing the feature similarity between $i$ and $j$ to be larger than that of $i$ and any other sample $k$ in the batch, if the label distance between $i$ and $k$ is larger than that of $i$ and $j$. Minimizing $l_{\text{RNC}}^{(i)}$ will align the orders of feature embeddings with their corresponding orders in the label space w.r.t. anchor $i$.

$\mathcal{L}_{\text{RNC}}$ is then enumerating over all $2N$ samples as anchors to enforce the entire feature embeddings ordered according to their orders in the label space:

$$\mathcal{L}_{\text{RNC}} = \frac{1}{2N} \sum_{i=1}^{2N} l_{\text{RNC}}^{(i)} = \frac{1}{2N} \sum_{i=1}^{2N} \frac{1}{2N-1} \sum_{j=1,\ j\neq i}^{2N} -\log \frac{\exp(\text{sim}(\boldsymbol{v}_i, \boldsymbol{v}_j)/\tau)}{\sum_{\boldsymbol{v}_k \in \mathcal{S}_{i,j}} \exp(\text{sim}(\boldsymbol{v}_i, \boldsymbol{v}_k)/\tau)}. \tag{3}$$

**Interpretation.** To exploit the inherent continuity underlying the labels, $\mathcal{L}_{\text{RNC}}$ ranks samples in a batch with respect to their label distances to the anchor. When contrasting the anchor to the sample in the batch that is *closest* in the label space, it enforces their similarity to be larger than all other samples in the batch. Similarly, when contrasting the anchor to the *second closest* sample in the batch, it enforces their similarity to be larger than only those samples that have a rank of three or higher in terms of distance to the anchor. This process is repeated for higher-rank samples (i.e., the third closest, fourth closest, etc.) and for all anchors in a batch.

**Feature Ordinality.** We further examine the impact of RNC on the ordinality of learned features. Fig. 3 visualizes the feature similarity matrices obtained from 2,000 randomly sampled data points in a real-world temperature regression task [7] for models trained using the vanilla $L_1$ loss and RNC. For clarity, the data points are sorted based on their ground-truth labels, with the expectation that the matrix values decrease progressively from the diagonal to the periphery. Notably, our method, RNC, exhibits a more discernible pattern compared to the $L_1$ loss. Furthermore, we calculate two quantitative metrics, the Spearman's rank correlation coefficient [41] and the Kendall rank correlation coefficient [24], between the label similarities and the feature similarities for both methods. The results in Table 1 confirm that the feature similarities learned by our method have a significantly higher correlation with label similarities than those by the $L_1$ loss.

Table 1: **Correlation between feature & label similarities.**

| | Spearman's $\rho^{\uparrow}$ | Kendall's $\tau^{\uparrow}$ |
|---|---|---|
| $L_1$ | 0.822 | 0.664 |
| RNC | **0.971** | **0.870** |

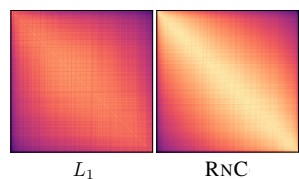

| $L_1$ | RNC |
|---|---|

Figure 3: **Feature similarity matrices sorted by labels.**

**Connections to Contrastive Learning.** The loss can also be explained in the context of positive and negative pairs in contrastive learning. Contrastive learning and SupCon are designed for classification tasks, where positive pairs consist of samples that belong to the same class or the same input image, while all other samples are considered as negatives. In regression, however, there are no distinct classes but rather continuous labels [44, 46]. Thus in $\mathcal{L}_{\text{RNC}}$, any two samples can be considered as a positive or negative pair depending on the context. For a given anchor sample $i$, *any* other sample $j$ in the same batch can be used to construct a positive pair with the corresponding negative samples set to all samples in the batch whose labels differ from $i$'s label by more than the label of $j$. Fig. 2(a) shows an example batch, and Fig. 2(b) shows two positive pairs and their corresponding negative pair(s).

## 4   Theoretical Analysis

In this section, we theoretically prove that optimizing $\mathcal{L}_{\text{RNC}}$ results in an ordered feature embedding that corresponds to the ordering of the labels. All proofs are in Appendix A.

**Notations.** Let $s_{i,j} := \text{sim}(\boldsymbol{v}_i, \boldsymbol{v}_j)/\tau$ and $d_{i,j} := d(\tilde{\boldsymbol{y}}_i, \tilde{\boldsymbol{y}}_j)$, $\forall i, j \in [2N]$. Let $D_{i,1} < D_{i,2} < \cdots < D_{i,M_i}$ be the sorted label distances starting from the $i$-th sample (i.e., $\text{sort}(\{d_{i,j} | j \in [2N] \setminus \{i\}\})$),

$\forall i \in [2N]$. Let $n_{i,m} := |\{j \mid d_{i,j} = D_{i,m}, \ j \in [2N]\backslash\{i\}\}|$ be the number of samples whose distance from the $i$-th sample equals $D_{i,m}, \forall i \in [2N], m \in [M_i]$.

First, to formalize the concept of ordered feature embedding according to the order in the label space, we introduce a property termed as $\delta$-*ordered* for a set of feature embeddings $\{v_l\}_{l \in [2N]}$.

**Definition 1** ($\delta$-ordered feature embeddings). For any $0 < \delta < 1$, the feature embeddings $\{v_l\}_{l \in [2N]}$ are $\delta$-ordered if $\forall i \in [2N], j, k \in [2N]\backslash\{i\}$,

$$
\begin{cases}
s_{i,j} > s_{i,k} + \dfrac{1}{\delta} & \text{if } d_{i,j} < d_{i,k} \\[2mm]
|s_{i,j} - s_{i,k}| < \delta & \text{if } d_{i,j} = d_{i,k} \\[2mm]
s_{i,j} < s_{i,k} - \dfrac{1}{\delta} & \text{if } d_{i,j} > d_{i,k}
\end{cases}
$$

Definition 1 implies that a set of feature embeddings that is $\delta$-ordered satisfies the following properties: ❶ For any $j$ and $k$ such that $d_{i,j} = d_{i,k}$, the difference between $s_{i,j}$ and $s_{i,k}$ is no more than $\delta$; and ❷ For any $j$ and $k$ such that $d_{i,j} < d_{i,k}$, the value of $s_{i,j}$ exceeds that of $s_{i,k}$ by at least $\frac{1}{\delta}$. Note that $\frac{1}{\delta} > \delta$, which indicates that the feature similarity gap between samples with different label distances to the anchor is always larger than that between samples with equal label distances to the anchor.

Next, we demonstrate that the batch of feature embeddings will be $\delta$-ordered as the optimization of $\mathcal{L}_{\text{RNC}}$ approaches its lower bound. In order to prove this, it is necessary to derive a tight lower bound for $\mathcal{L}_{\text{RNC}}$. Let $L^\star := \frac{1}{2N(2N-1)} \sum_{i=1}^{2N} \sum_{m=1}^{M_i} n_{i,m} \log n_{i,m}$, we have:

**Theorem 1** (Lower bound of $\mathcal{L}_{\text{RNC}}$). *$L^\star$ is a lower bound of $\mathcal{L}_{\text{RNC}}$, i.e., $\mathcal{L}_{\text{RNC}} > L^\star$.*

**Theorem 2** (Lower bound tightness). *For any $\epsilon > 0$, there exists a set of feature embeddings such that $\mathcal{L}_{\text{RNC}} < L^\star + \epsilon$.*

The above theorems verify the lower bound of $\mathcal{L}_{\text{RNC}}$ as well as its tightness. Given that $\mathcal{L}_{\text{RNC}}$ can approach its lower bound $L^\star$ arbitrarily, we demonstrate that the feature embeddings will be $\delta$-ordered when $\mathcal{L}_{\text{RNC}}$ is sufficiently close to $L^\star$ for any $0 < \delta < 1$.

**Theorem 3** (Main theorem). *For any $0 < \delta < 1$, there exist $\epsilon > 0$, such that if $\mathcal{L}_{\text{RNC}} < L^\star + \epsilon$, then the feature embeddings are $\delta$-ordered.*

**From Batch to Entire Feature Space.** Now we have examined the property of a batch of feature embeddings optimized using $\mathcal{L}_{\text{RNC}}$. However, what will be the final outcome for the *entire* feature space when $\mathcal{L}_{\text{RNC}}$ is optimized? In fact, if any batch of feature embeddings is optimized to achieve a low enough loss such that it is $\delta$-ordered, the entire feature embedding will also be $\delta$-ordered. This is because any triplet $(i, j, k)$ in the entire feature embeddings is certain to appear in some batch, thus their feature embeddings $(v_i, v_j, v_k)$ will satisfy the condition in Definition 1. Nevertheless, to achieve $\delta$-ordered features for the entire feature embeddings, do we need to optimize all batches to achieve a sufficiently low loss? The answer is no. Optimizing every batch is not only unnecessary, but also practically infeasible. In fact, one should consider the training process as a cohesive whole, which is effectively optimizing the *expectation* of the loss over all possible random batches. Then, the Markov's inequality [18] guarantees that when the expectation of the loss is optimized to be sufficiently low, the loss on *any* batch will be low enough with a high probability.

**Connections to Final Performance.** Suppose we have a $\delta$-ordered feature embedding, how can it help to boost the *final performance* of a regression task? In Appendix B, we present an analysis based on Rademacher Complexity [39] to prove that a $\delta$-ordered feature embedding results in a better generalization bound. To put it intuitively, fitting an ordered feature embedding reduces the complexity of the regressor, which enables better generalization ability from training to testing, and ultimately leads to the final performance gain. Relatedly, we note that the enhanced generalization ability is further empirically verified in Sec. 5. Specifically, if not constrained, the learned feature embeddings could capture spurious or easy-to-learn features that are not generalizable to the real continuous targets. Moreover, such property also leads to better robustness to data corruptions, better resilience to reduced training data, and better generalization to unseen targets.

Table 2: **Comparison and compatibility to end-to-end regression methods.** RNC is compatible to end-to-end regression methods, and consistently improves the performance over all datasets.

| Metrics | AgeDB | | TUAB | | MPIIFaceGaze | | SkyFinder | |
|---|---|---|---|---|---|---|---|---|
| | MAE$\downarrow$ | R$^{2\uparrow}$ | MAE$\downarrow$ | R$^{2\uparrow}$ | Angular$\downarrow$ | R$^{2\uparrow}$ | MAE$\downarrow$ | R$^{2\uparrow}$ |
| $L_1$ | 6.63 | 0.828 | 7.46 | 0.655 | 5.97 | 0.744 | 2.95 | 0.860 |
| **RNC($L_1$)** | **6.14** (+0.49) | **0.850** (+0.022) | **6.97** (+0.49) | **0.697** (+0.042) | **5.27** (+0.70) | **0.815** (+0.071) | **2.86** (+0.09) | **0.869** (+0.009) |
| MSE | 6.57 | 0.828 | 8.06 | 0.585 | 6.02 | 0.747 | 3.08 | 0.851 |
| **RNC(MSE)** | **6.19** (+0.38) | **0.849** (+0.021) | **7.05** (+1.01) | **0.692** (+0.107) | **5.35** (+0.67) | **0.802** (+0.055) | **2.86** (+0.22) | **0.869** (+0.018) |
| HUBER | 6.54 | 0.828 | 7.59 | 0.637 | 6.34 | 0.709 | 2.92 | 0.860 |
| **RNC(HUBER)** | **6.15** (+0.39) | **0.850** (+0.022) | **6.99** (+0.60) | **0.696** (+0.059) | **5.15** (+1.19) | **0.830** (+0.121) | **2.86** (+0.06) | **0.869** (+0.009) |
| DEX [36] | 7.29 | 0.787 | 8.01 | 0.537 | 5.72 | 0.776 | 3.58 | 0.778 |
| **RNC(DEX)** | **6.43** (+0.86) | **0.836** (+0.049) | **7.23** (+0.78) | **0.646** (+0.109) | **5.14** (+0.58) | **0.805** (+0.029) | **2.88** (+0.70) | **0.865** (+0.087) |
| DLDL-v2 [14] | 6.60 | 0.827 | 7.91 | 0.560 | 5.47 | 0.799 | 2.99 | 0.856 |
| **RNC(DLDL-v2)** | **6.32** (+0.28) | **0.844** (+0.017) | **6.91** (+1.00) | **0.697** (+0.137) | **5.16** (+0.31) | **0.802** (+0.003) | **2.85** (+0.14) | **0.869** (+0.013) |
| OR [33] | 6.40 | 0.830 | 7.36 | 0.646 | 5.86 | 0.770 | 2.92 | 0.861 |
| **RNC(OR)** | **6.34** (+0.06) | **0.843** (+0.013) | **7.01** (+0.35) | **0.688** (+0.042) | **5.13** (+0.73) | **0.825** (+0.055) | **2.86** (+0.06) | **0.867** (+0.006) |
| CORN [40] | 6.72 | 0.811 | 8.11 | 0.597 | 5.88 | 0.762 | 3.24 | 0.819 |
| **RNC(CORN)** | **6.44** (+0.28) | **0.838** (+0.027) | **7.22** (+0.89) | **0.663** (+0.066) | **5.18** (+0.70) | **0.820** (+0.058) | **2.89** (+0.35) | **0.862** (+0.043) |

## 5 Experiments

**Datasets.** We perform extensive experiments on five regression datasets that span different tasks and domains, including computer vision, human-computer interaction, and healthcare. Complete descriptions of each dataset and example inputs are in Table 7, Appendix C and D.

- AgeDB *(Age)* [32, 44] is a dataset for predicting age from face images, containing 16,488 in-the-wild images of celebrities and the corresponding age labels.
- IMDB-WIKI *(Age)* [36, 44] is a dataset for age prediction from face images, which contains 523,051 celebrity images and the corresponding age labels. We use this dataset only for the analysis.
- TUAB *(Brain-Age)* [34, 11] aims for brain-age estimation from EEG resting-state signals, with 1,385 21-channel EEG signals sampled at 200Hz from individuals with age from 0 to 95.
- MPIIFaceGaze *(Gaze Direction)* [51, 52] contains 213,659 face images collected from 15 participants during natural everyday laptop use. The task is to predict the gaze direction.
- SkyFinder *(Temperature)* [31, 7] contains 35,417 images captured by 44 outdoor webcam cameras for in-the-wild temperature prediction.

**Metrics.** We report two distinct metrics for model evaluation: ❶ Prediction error, and ❷ Coefficient of determination ($R^2$). Prediction error provides practical insight for interpretation, while $R^2$ quantifies the difficulty of the task by measuring the extent to which the model outperforms a dummy regressor that always predicts the mean value of the training labels. We report the mean absolute error (MAE) for age, brain-age, and temperature prediction, and the angular error for gaze direction estimation.

**Experiment Settings.** We employ ResNet-18 [19] as the main backbone for AgeDB, IMDB-WIKI, MPIIFaceGaze, and SkyFinder. In addition, we confirm that the results are consistent with other backbones (e.g., ResNet-50), and report related results in Appendix G.1. For TUAB, a 24-layer 1D ResNet [19] is used as the backbone model to process the EEG signals, with a linear regressor acting as the predictor. We fix data augmentations to be the same for all methods, where the details are reported in Appendix D. Negative $L_2$ norm (i.e., $-\|v_i - v_j\|_2$) is used as the feature similarity measure in $\mathcal{L}_{\text{RNC}}$, with $L_1$ distance as the label distance measure for AgeDB, IMDB-WIKI, SkyFinder, and TUAB, and angular distance as the label distance measure for MPIIFaceGaze. We provide complete experimental settings and hyper-parameter choices in Appendix F.

### 5.1 Main Results

**Comparison and compatibility to end-to-end regression methods.** As explained earlier, our model learns a regression-suitable representation that can be used by any end-to-end regression methods. Thus, we implement **seven** generic regression methods: $L_1$, MSE, and HUBER [44] ask the model to directly predict the target and utilize an error-based loss function; DEX [36] and DLDL-v2 [14] divide the regression range of each label dimension into pre-defined bins and learn the probability

Table 3: **Comparisons to state-of-the-art representation & regression learning methods.** MAE$^\downarrow$ is used as the metric for `AgeDB`, `TUAB`, and `SkyFinder`, and Angular Error$^\downarrow$ is used for `MPIIFaceGaze`. $L_1$ loss is employed as the default regression loss if not specified. RNC surpasses state-of-the-art methods on all datasets.

| Method | AgeDB | TUAB | MPIIFaceGaze | SkyFinder |
|---|---|---|---|---|
| *Representation learning methods (Linear Probing):* | | | | |
| SIMCLR [4] | 9.59 | 11.01 | 9.43 | 4.70 |
| DINO [3] | 10.26 | 11.62 | 11.92 | 5.63 |
| SUPCON [25] | 8.13 | 8.47 | 9.27 | 3.97 |
| *Representation learning methods (Fine-tuning):* | | | | |
| SIMCLR [4] | 6.57 | 7.57 | 5.50 | 2.93 |
| DINO [3] | 6.61 | 7.58 | 5.80 | 2.98 |
| SUPCON [25] | 6.55 | 7.41 | 5.54 | 2.95 |
| *Regression learning methods:* | | | | |
| $L_1$ | 6.63 | 7.46 | 5.97 | 2.95 |
| LDS+FDS [44] | 6.45 | – | – | – |
| L2CS-NET [1] | – | – | 5.45 | – |
| LDE [7] | – | – | – | 2.92 |
| RANKSIM [17] | 6.51 | 7.33 | 5.70 | 2.94 |
| ORDINAL ENTROPY [50] | 6.47 | 7.28 | – | 2.94 |
| **RNC($L_1$)** | **6.14** | **6.97** | **5.27** | **2.86** |
| GAINS | **+0.31** | **+0.31** | **+0.18** | **+0.06** |

distribution over them; OR [33] and CORN [40] design multiple ordered thresholds for each label dimension and learn a binary classifier for each threshold. Further details of these baselines are included in Appendix E.1. In our comparison, we first train the encoder with the proposed $\mathcal{L}_{\text{RNC}}$. We then freeze the encoder and train a predictor on top of it using each of the baseline methods. The original baseline without the RNC representation is then compared to that with RNC. For instance, we denote the end-to-end training of the encoder and predictor using $L_1$ loss as $L_1$, while RNC($L_1$) denotes training the encoder with $\mathcal{L}_{\text{RNC}}$ and subsequently training the predictor using $L_1$ loss.

Table 2 summarizes the evaluation results on `AgeDB`, `TUAB`, `MPIIFaceGaze` and `SkyFinder`. Green numbers highlight the performance gains by using RNC representation. The standard deviations of the best results on each dataset are reported in Appendix G.2. As the table indicates, RNC consistently achieves the best performance on both metrics across all datasets. Moreover, incorporating RNC to learn the representation consistently reduces the prediction error of all baselines by **5.8%**, **9.3%**, **11.7%**, and **7.0%** on average on `AgeDB`, `TUAB`, `MPIIFaceGaze`, and `SkyFinder`, respectively.

**Comparison to state-of-the-art representation learning methods.** We further compare RNC with state-of-the-art representation learning methods, SIMCLR [4], DINO [3], and SUPCON [25], under both *linear probing* and *fine-tuning* schemes to evaluate their learned representations for regression tasks. Full details are in Appendix E.2. Note that SIMCLR and DINO do not use label information while SUPCON uses label information for training the encoder. The predictor is trained with $L_1$ loss. Table 3 demonstrates that RNC outperforms all other methods across all datasets. Note that some of the representation learning schemes even underperform the vanilla $L_1$ method, which further verifies that the performance gain of RNC stems from our proposed loss rather than the pre-training scheme.

**Comparison to state-of-the-art regression learning methods.** We also compare RNC with state-of-the-art regression learning schemes, including methods that perform regularization in the feature space [44, 17, 50] and methods that adopt dataset-specific techniques [1, 7]. We provide the details in Appendix E.3. $L_1$ loss is employed as the default regression loss for all methods if not specified. The results are presented in Table 3, with a dash (−) indicating that the method is not applicable to the dataset. We observe that RNC achieves state-of-the-art performance across all datasets.

## 5.2 Analysis

**Robustness to Data Corruptions.** Deep neural networks are widely acknowledged to be vulnerable to out-of-distribution data and various forms of corruption, such as noise, blur, and color distortions [21]. To analyze the robustness of RNC, we generate corruptions on the `AgeDB` test set using the corruption process from the ImageNet-C benchmark [21], incorporating 19 distinct types of corruption at varying severity levels. We compare RNC($L_1$) with $L_1$ by training both models on the original `AgeDB` training set, but directly testing them on the corrupted test set across all severity levels. The

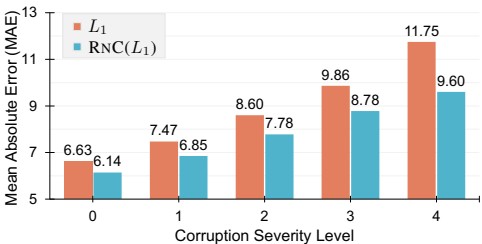
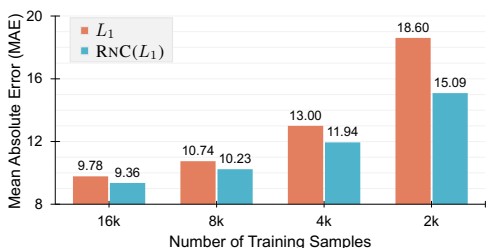

Figure 4: **RNC is more robust to data corruptions.** We show MAE$^\downarrow$ as a function of corruption severity on `AgeDB` test set.

Figure 5: **RNC is more resilient to reduced training data.** We show MAE$^\downarrow$ as a function of the number of training samples on `IMDB-WIKI`.

Table 4: **Transfer learning results.** RNC delivers better performance than $L_1$ in all scenarios.

| | AgeDB → IMDB-WIKI (subsampled, 2k) | | | | IMDB-WIKI (subsampled, 32k) → AgeDB | | | |
| | *Linear Probing* | | *Fine-tuning* | | *Linear Probing* | | *Fine-tuning* | |
| Metrics | MAE$^\downarrow$ | R$^{2\uparrow}$ | MAE$^\downarrow$ | R$^{2\uparrow}$ | MAE$^\downarrow$ | R$^{2\uparrow}$ | MAE$^\downarrow$ | R$^{2\uparrow}$ |
|---|---|---|---|---|---|---|---|---|
| $L_1$ | 12.25 | 0.496 | 11.57 | 0.528 | 7.36 | 0.801 | 6.36 | 0.848 |
| **RNC**($L_1$) | **11.12** (+1.13) | **0.556** (+0.060) | **11.09** (+0.48) | **0.546** (+0.018) | **7.06** (+0.30) | **0.812** (+0.011) | **6.13** (+0.23) | **0.850** (+0.002) |

reported results are averaged over all types of corruptions. Fig. 4 illustrates the results, where the representation learned by RNC is more robust to unforeseen data corruptions, with consistently less performance degradation when corruption severity increases.

**Resilience to Reduced Training Data.** The availability of massive training datasets has played an important role in the success of modern deep learning. However, in many real-world scenarios, the cost and time involved in labeling large training sets make it infeasible to do so. As a result, there is a need to enhance model resilience to limited training data. To investigate this, we subsample `IMDB-WIKI` to generate training sets of varying sizes and compare the performance of RNC($L_1$) and $L_1$. As Fig. 5 confirms, RNC is more robust to reduced training data and displays less performance degradation as the number of training samples decreases.

**Transfer Learning.** We evaluate whether the learned representations by RNC are transferable across datasets. To do so, we first pre-train the feature encoder on a large dataset, and subsequently utilize either *linear probing* (fixed encoder) or *fine-tuning* to learn a predictor on a small dataset (which shares the same prediction task). We investigate two scenarios: ❶ Transferring from `AgeDB` which contains ∼12K samples to a subsampled `IMDB-WIKI` of 2K samples, and ❷ Transferring from another subsampled `IMDB-WIKI` of 32K samples to `AgeDB`. As shown in Table 4, RNC($L_1$) exhibits consistent and superior performance compared to $L_1$ in both linear probing and fine-tuning settings for both of the aforementioned scenarios.

**Robustness to Spurious Targets.** We show that RNC is able to deal with spurious targets that arise in data [46], while existing regression learning methods often fail to learn generalizable features. Specifically, the `SkyFinder` dataset naturally has a spurious target: different *webcams* (with distinct locations). Therefore, we show in Fig. 6 the UMAP [30] visualization of the learned features obtained from both $L_1$ and RNC, using data from 10 webcams in `SkyFinder`. The first column of the figure is colored by the ground-truth target (temperature), while the second column is colored by the 10 different webcams. Our results demonstrate that the representation learned by $L_1$ is clustered by webcam, indicating that it is prone to capturing easy-to-learn features. In contrast, RNC learns the underlying continuous temperature information even in the presence of strong spurious targets, confirming its ability to learn robust representations that generalize.

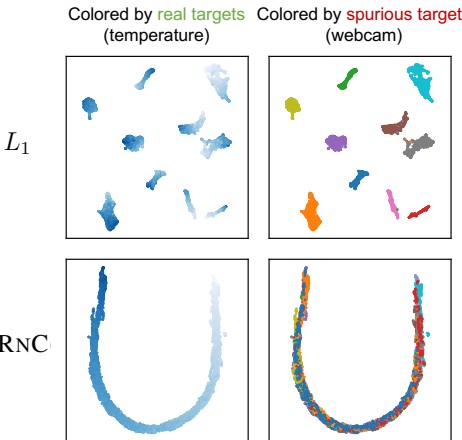

Figure 6: **Robustness to spurious targets.** RNC is able to capture underlying continuous targets & learn robust representations that generalize.

Table 6: **Ablation studies for RNC.** All experiments are performed on the `AgeDB` dataset and $L_1$ is used as the default loss for training the predictor. Default settings used in the main experiments are marked in gray.

(a) Number of Positives $K$

| | MAE$^\downarrow$ | R$^{2\uparrow}$ |
|---|---|---|
| 128 | 6.46 | 0.828 |
| 256 | 6.43 | 0.833 |
| 384 | 6.29 | 0.845 |
| 511 | **6.14** | **0.850** |

(b) Feature Similarity Measure

| | MAE$^\downarrow$ | R$^{2\uparrow}$ |
|---|---|---|
| cosine | 6.51 | 0.836 |
| negative $L_1$ norm | 6.25 | 0.842 |
| negative $L_2$ norm | **6.14** | **0.850** |

(c) Training Scheme

| | MAE$^\downarrow$ | R$^{2\uparrow}$ |
|---|---|---|
| linear probing | **6.14** | **0.850** |
| fine-tuning | 6.36 | 0.844 |
| regularization | 6.42 | 0.833 |

**Generalization to Unseen Targets.** In practical regression tasks, it is frequently encountered that some targets are unseen during training. To investigate **zero-shot** generalization to unseen targets, following [44], we curate two subsets of `IMDB-WIKI` that contain unseen targets during training, while keeping the test set uniformly distributed across the target range. Table 5 shows the label distributions, where regions of unseen targets are marked with pink shading and those of seen targets are marked with blue shading. The first training set has a bi-modal Gaussian distribution, while the second one exhibits a tri-modal Gaussian distribution over the target space. The results confirm that RNC($L_1$) outperforms $L_1$ by a larger margin on the unseen targets without sacrificing the performance on the seen targets.

Table 5: **Zero-shot generalization to unseen targets.** We create two `IMDB-WIKI` training sets with missing targets, and keep test sets uniformly distributed across the target range. MAE$^\downarrow$ is used as the metric.

| Label Distribution | Method | All | Seen | Unseen |
|---|---|---|---|---|
|  | $L_1$ | 12.53 | 10.82 | 18.40 |
| | RNC($L_1$) | **11.69** | **10.46** | **15.92** |
| | | (+0.84) | (+0.36) | (+2.48) |
|  | $L_1$ | 11.94 | 10.43 | 14.98 |
| | RNC($L_1$) | **10.88** | **9.78** | **13.08** |
| | | (+1.06) | (+0.64) | (+1.90) |

## 5.3 Ablation Studies

**Ablation on Number of Positives.** Recall that in $\mathcal{L}_{\text{RNC}}$, all samples in the batch will be treated as the positive for each anchor. Here, we conduct an ablation study on only considering the first $K$ closest samples to the anchor as positive. Table 6a shows the results on `AgeDB` for different $K$, where $K = 511$ represents the scenario where all samples are considered as positive (default batch size $N = 256$). The experiments reveal that larger values of $K$ lead to better performance, which aligns with the intuition behind the design of $\mathcal{L}_{\text{RNC}}$: each contrastive term ensures that a group of orders related to the positive sample is maintained. Specifically, it guarantees that all samples that have a larger label distance from the anchor than the positive sample are farther from the anchor than the positive sample in the feature space. Only when all samples are treated as positive and their corresponding groups of orders are preserved can the order in the feature space be fully guaranteed.

**Ablation on Similarity Measure.** Table 6b shows the performance of using different feature similarity measures $\text{sim}(\cdot, \cdot)$. Compared to cosine similarity, both the negative $L_1$ norm and $L_2$ norm produce significantly better results. This improvement can be attributed to the fact that cosine similarity only captures the directions of feature vectors, while the negative $L_1$ or $L_2$ norm takes both the direction and magnitude of the vectors into account. This finding highlights the potential differences between representation learning for classification and regression tasks, where the standard practice for classification is to use cosine similarity [4, 25], while our findings suggest the superiority of $L_1$ and $L_2$ norm for regression-based representation learning.

**Ablation on Training Scheme.** There are typically three schemes to train the encoder: ❶ *Linear probing*, which first trains the feature encoder using the representation learning loss, then fixes the encoder and trains a linear regressor on top of it using a regression loss; ❷ *Fine-tuning*. which first trains the feature encoder using the representation learning loss, then fine-tunes the entire model using a regression loss; and ❸ *Regularization*, which trains the entire model while jointly optimizing the representation learning & the regression loss. Table 6c shows the results for the three schemes using $\mathcal{L}_{\text{RNC}}$ as the representation learning loss and $L_1$ as the regression loss. All three schemes can improve performance over using $L_1$ loss alone. Further, unlike classification problems where fine-tuning often delivers the best performance, freezing the feature encoder yields the best outcome in regression. This is because, in the case of regression, back-propagating the $L_1$ loss to the representation can disrupt the order in the embedding space learned by $\mathcal{L}_{\text{RNC}}$, leading to inferior performance.

### 5.4 Further Discussions

**Does RNC rely on data augmentation for superior performance (Appendix G.3)?** Table 10 confirms that RNC surpasses the baseline, *with* or *without* data augmentation. In fact, unlike typical (unsupervised) contrastive learning techniques that rely on data augmentation for distinguishing the augmented views, data augmentation is *not essential* for RNC. This is because RNC contrasts samples according to label distance rather than the identity. The role of data augmentation for RNC is similar to its role in typical regression learning, which is to enhance model generalization.

**Does the benefit of RNC come from the two-stage training scheme (Appendix G.4)?** We confirm in Table 11 that training a predictor on top of the representation learned by the competing regression baselines does not improve performance. In fact, it can even be *detrimental* to performance. This finding validates that the benefit of RNC is due to the proposed $\mathcal{L}_{\text{RNC}}$ and not the training scheme. The generic regression losses are not designed for representation learning and are computed based on the final model predictions, which fails to guarantee the final learned representation.

**Computational cost of RNC (Appendix G.5)?** We verify in Table 12 that the training time of RNC is comparable to standard contrastive learning methods (e.g., SUPCON [25]), indicating that RNC offers similar training efficiency, while achieving notably better performance for regression tasks.

## 6 Broader Impacts and Limitations

**Broader Impacts.** We introduce a novel framework designed to enhance the performance of generic deep regression learning. We believe this will significantly benefit regression tasks across various real-world applications. Nonetheless, several potential risks warrant discussion. First, when the framework is employed to regress sensitive personal attributes such as intellectual capabilities, health status, or financial standing from human data, there's a danger it might reinforce or even introduce new forms of bias. Utilizing the method in these contexts could inadvertently justify discrimination or the negative targeting of specific groups. Second, in contrast to handcrafted features which are grounded in physical interpretations, the feature representations our framework learns can be opaque. This makes it difficult to understand and rationalize the model's predictions, particularly when trying to determine if any biases exist. Third, when our method is trained on datasets that do not have a balanced representation of minority groups, there's no assurance of its performance on these groups being reliable. It is essential to recognize that these ethical concerns apply to deep regression models at large, not solely our method. However, the continuous nature of our representation which facilitates interpolation and extrapolation might inadvertently make it more tempting to justify such unethical applications. Anyone seeking to implement or use our proposed method should be mindful of these concerns. Both our specific method and deep regression models, in general, should be used cautiously to avoid situations where their deployment might contribute to unethical outcomes or interpretations.

**Limitations.** Our proposed method presents some limitations. Firstly, the technique cannot discern spurious or incidental correlations between the input and the target within the dataset. As outlined in the Broader Impact section, this could result in incorrect conclusions potentially promoting discrimination or unjust treatment when utilized to deduce personal attributes. Future research should delve deeper into the ethical dimensions of this issue and explore strategies to ensure ethical regression learning. A second limitation is that our evaluation primarily focuses on general regression accuracy metrics (e.g., MAE) without considering potential disparities when evaluating specific subgroups (e.g., minority groups). Given that a regression model's performance can vary across demographic segments, subgroup analysis is an avenue that warrants exploration in subsequent studies. Lastly, our approach learns continuous representations by contrasting samples against one another based on their ranking in the target space, necessitating label information. To adapt it for representation learning with unlabeled data, our framework will need some modifications, which we reserve for future work.

## 7 Conclusion

We present Rank-N-Contrast (RNC), a framework that learns continuous representations for regression by ranking samples according to their labels and then contrasting them against each other based on their relative rankings. Extensive experiments on different datasets over various real-world tasks verify the superior performance of RNC, highlighting its intriguing properties such as better data efficiency, robustness to corruptions and spurious targets, and generalization to unseen targets.

**Acknowledgements.** We are grateful to Hao He for his invaluable assistance with the theoretical analysis in the paper. We thank the members of the NETMIT group for their constructive feedback on the draft of this paper. We extend our appreciation to the anonymous reviewers for their insightful comments and suggestions that greatly helped in improving the quality of the paper. Lastly, we acknowledge the generous support from the GIST-MIT program, which funded this project.

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

# A Proofs

## A.1 Proof of Theorem 1

Recall that in Eqn. (3) we defined $\mathcal{L}_{\text{RNC}} := \frac{1}{2N} \sum_{i=1}^{2N} \frac{1}{2N-1} \sum_{j=1,\ j\neq i}^{2N} -\log \frac{\exp(s_{i,j})}{\sum_{\boldsymbol{v}_k \in \mathcal{S}_{i,j}} \exp(s_{i,k})}$, where $\mathcal{S}_{i,j} := \{\boldsymbol{v}_k \mid k \neq i, d_{i,k} \geq d_{i,j}\}$. We rewrite it as

$$
\begin{aligned}
\mathcal{L}_{\text{RNC}} = &- \frac{1}{2N(2N-1)} \sum_{i=1}^{2N} \sum_{j\in[2N]\backslash\{i\}} \log \frac{\exp(s_{i,j})}{\sum\limits_{k\in[2N]\backslash\{i\},\, d_{i,k}\geq d_{i,j}} \exp(s_{i,k})} \\
= &- \frac{1}{2N(2N-1)} \sum_{i=1}^{2N} \sum_{m=1}^{M_i} \sum_{j\in[2N]\backslash\{i\},\, d_{i,j}=D_{i,m}} \log \frac{\exp(s_{i,j})}{\sum\limits_{k\in[2N]\backslash\{i\},\, d_{i,k}\geq D_{i,m}} \exp(s_{i,k})} \\
= &- \frac{1}{2N(2N-1)} \sum_{i=1}^{2N} \sum_{m=1}^{M_i} \sum_{j\in[2N]\backslash\{i\},\, d_{i,j}=D_{i,m}} \log \frac{\exp(s_{i,j})}{\sum\limits_{k\in[2N]\backslash\{i\},\, d_{i,k}=D_{i,m}} \exp(s_{i,k})} \\
&+ \frac{1}{2N(2N-1)} \sum_{i=1}^{2N} \sum_{m=1}^{M_i} \sum_{j\in[2N]\backslash\{i\},\, d_{i,j}=D_{i,m}} \log \left( 1 + \frac{\sum\limits_{k\in[2N]\backslash\{i\},\, d_{i,k}>D_{i,m}} \exp(s_{i,k}-s_{i,j})}{\sum\limits_{k\in[2N]\backslash\{i\},\, d_{i,k}=D_{i,m}} \exp(s_{i,k}-s_{i,j})} \right) \\
> &- \frac{1}{2N(2N-1)} \sum_{i=1}^{2N} \sum_{m=1}^{M_i} \sum_{j\in[2N]\backslash\{i\},\, d_{i,j}=D_{i,m}} \log \frac{\exp(s_{i,j})}{\sum\limits_{k\in[2N]\backslash\{i\},\, d_{i,k}=D_{i,m}} \exp(s_{i,k})}.
\end{aligned}
\tag{4}
$$

$\forall i \in [2N], m \in [M_i]$, from Jensen's Inequality we have

$$
\begin{aligned}
&- \sum_{j\in[2N]\backslash\{i\},\, d_{i,j}=D_{i,m}} \log \frac{\exp(s_{i,j})}{\sum\limits_{k\in[2N]\backslash\{i\},\, d_{i,k}=D_{i,m}} \exp(s_{i,k})} \\
&\geq -n_{i,m} \log \left( \frac{1}{n_{i,m}} \sum_{j\in[2N]\backslash\{i\},\, d_{i,j}=D_{i,m}} \frac{\exp(s_{i,j})}{\sum\limits_{k\in[2N]\backslash\{i\},\, d_{i,k}=D_{i,m}} \exp(s_{i,k})} \right) = n_{i,m} \log n_{i,m}.
\end{aligned}
\tag{5}
$$

Thus, by plugging Eqn. (5) into Eqn. (4), we have

$$
\mathcal{L}_{\text{RNC}} > \frac{1}{2N(2N-1)} \sum_{i=1}^{2N} \sum_{m=1}^{M_i} n_{i,m} \log n_{i,m} = L^\star.
\tag{6}
$$

## A.2 Proof of Theorem 2

We will show $\forall \epsilon > 0$, there is a set of feature embeddings where

$$
\begin{cases}
s_{i,j} > s_{i,k} + \gamma & \text{if } d_{i,j} < d_{i,k} \\
s_{i,j} = s_{i,k} & \text{if } d_{i,j} = d_{i,k}
\end{cases}
$$

and $\gamma := \log \frac{2N}{\min\limits_{i\in[2N], m\in[M_i]} n_{i,m}\epsilon}$, $\forall i \in [2N], j,k \in [2N]\backslash\{i\}$, such that $\mathcal{L}_{\text{RNC}} < L^\star + \epsilon$.

For such a set of feature embeddings, $\forall i \in [2N], m \in [M_i], j \in \{j \in [2N]\backslash\{i\} \mid d_{i,j} = D_{i,m}\}$,

$$
- \log \frac{\exp(s_{i,j})}{\sum\limits_{k\in[2N]\backslash\{i\},\, d_{i,k}=D_{i,m}} \exp(s_{i,k})} = \log n_{i,m}
\tag{7}
$$

since $s_{i,k} = s_{i,j}$ for all $k$ such that $d_{i,k} = D_{i,m} = d_{i,j}$, and

$$\log\left(1 + \frac{\sum\limits_{k\in[2N]\setminus\{i\},\, d_{i,k}>D_{i,m}} \exp(s_{i,k} - s_{i,j})}{\sum\limits_{k\in[2N]\setminus\{i\},\, d_{i,k}=D_{i,m}} \exp(s_{i,k} - s_{i,j})}\right) \tag{8}$$

$$< \log\left(1 + \frac{2N\exp(-\gamma)}{n_{i,m}}\right) < \frac{2N\exp(-\gamma)}{n_{i,m}} \leq \epsilon$$

since $s_{i,k} - s_{i,j} < -\gamma$ for all $k$ such that $d_{i,k} > D_{i,m} = d_{i,j}$ and $s_{i,k} - s_{i,j} = 0$ for all $k$ such that $d_{i,k} = D_{i,m} = d_{i,j}$.

From Eqn. (4) we have

$$\mathcal{L}_{\text{RNC}} = -\frac{1}{2N(2N-1)} \sum_{i=1}^{2N}\sum_{m=1}^{M_i}\sum_{j\in[2N]\setminus\{i\},\, d_{i,j}=D_{i,m}} \log \frac{\exp(s_{i,j})}{\sum\limits_{k\in[2N]\setminus\{i\},\, d_{i,k}=D_{i,m}} \exp(s_{i,k})}$$

$$+ \frac{1}{2N(2N-1)} \sum_{i=1}^{2N}\sum_{m=1}^{M_i}\sum_{j\in[2N]\setminus\{i\},\, d_{i,j}=D_{i,m}} \log\left(1 + \frac{\sum\limits_{k\in[2N]\setminus\{i\},\, d_{i,k}>D_{i,m}} \exp(s_{i,k} - s_{i,j})}{\sum\limits_{k\in[2N]\setminus\{i\},\, d_{i,k}=D_{i,m}} \exp(s_{i,k} - s_{i,j})}\right), \tag{9}$$

By plugging Eqn. (7) and Eqn. (8) into Eqn. (9) we have

$$\mathcal{L}_{\text{RNC}} < \frac{1}{2N(2N-1)} \sum_{i=1}^{2N}\sum_{m=1}^{M_i} n_{i,m}\log n_{i,m} + \epsilon = L^\star + \epsilon \tag{10}$$

## A.3  Proof of Theorem 3

We will show $\forall 0 < \delta < 1$, there is a

$$\epsilon = \frac{1}{2N(2N-1)} \min\left(\min_{i\in[2N],m\in[M_i]} \log\left(1 + \frac{1}{n_{i,m}\exp(\delta + \frac{1}{\delta})}\right), 2\log\frac{1+\exp(\delta)}{2} - \delta\right) > 0,$$

such that when $\mathcal{L}_{\text{RNC}} < L^\star + \epsilon$, the feature embeddings are $\delta$-ordered.

We first show that $|s_{i,j} - s_{i,k}| < \delta$ if $d_{i,j} = d_{i,k}$, $\forall i \in [2N]$, $j, k \in [2N]\setminus\{i\}$ when $\mathcal{L}_{\text{RNC}} < L^\star + \epsilon$.

From Eqn. (4) we have

$$\mathcal{L}_{\text{RNC}} > -\frac{1}{2N(2N-1)} \sum_{i=1}^{2N}\sum_{m=1}^{M_i}\sum_{j\in[2N]\setminus\{i\},\, d_{i,j}=D_{i,m}} \log \frac{\exp(s_{i,j})}{\sum\limits_{k\in[2N]\setminus\{i\},\, d_{i,k}=D_{i,m}} \exp(s_{i,k})}. \tag{11}$$

Let $p_{i,m} := \underset{j\in[2N]\setminus\{i\},\, d_{i,j}=D_{i,m}}{\arg\min} s_{i,j}$, $q_{i,m} := \underset{j\in[2N]\setminus\{i\},\, d_{i,j}=D_{i,m}}{\arg\max} s_{i,j}$, $\zeta_{i,m} := s_{i,p_{i,m}}$, $\eta_{i,m} := s_{i,q_{i,m}} - s_{i,p_{i,m}}$, $\forall i \in [2N], m \in [M_i]$, by splitting out the maximum term and the minimum term we have

$$\mathcal{L}_{\text{RNC}} > -\frac{1}{2N(2N-1)} \sum_{i=1}^{2N}\sum_{m=1}^{M_i}\left\{\log\frac{\exp(\zeta_{i,m})}{\sum\limits_{k\in[2N]\setminus\{i\},\, d_{i,k}=D_{i,m}} \exp(s_{i,k})}\right.$$

$$\left. + \log\frac{\exp(\zeta_{i,m} + \eta_{i,m})}{\sum\limits_{k\in[2N]\setminus\{i\},\, d_{i,k}=D_{i,m}} \exp(s_{i,k})} + \log\frac{\exp\left(\sum\limits_{j\in[2N]\setminus\{i,p_{i,m},q_{i,m}\},d_{i,j}=D_{i,m}} s_{i,j}\right)}{\left(\sum\limits_{k\in[2N]\setminus\{i\},\, d_{i,k}=D_{i,m}} \exp(s_{i,k})\right)^{n_{i,m}-2}}\right\}. \tag{12}$$

Let $\theta_{i,m} := \frac{1}{n_{i,m}-2} \sum\limits_{j\in[2N]\setminus\{i,p_{i,m},q_{i,m}\},d_{i,j}=D_{i,m}} \exp(s_{i,j} - \zeta_{i,m})$, we have

$$-\log\frac{\exp(\zeta_{i,m})}{\sum\limits_{k\in[2N]\setminus\{i\},\, d_{i,k}=D_{i,m}} \exp(s_{i,k})} = \log(1 + \exp(\eta_{i,m}) + (n_{i,m}-2)\theta_{i,m}) \tag{13}$$

and

$$-\log \frac{\exp(\zeta_{i,m} + \eta_{i,m})}{\sum\limits_{k\in[2N]\setminus\{i\},\,d_{i,k}=D_{i,m}} \exp(s_{i,k})} = \log(1 + \exp(\eta_{i,m}) + (n_{i,m} - 2)\theta_{i,m}) - \eta_{i,m}. \quad (14)$$

Then, from Jensen's inequality, we know

$$\exp\left(\sum_{j\in[2N]\setminus\{i,p_{i,m},q_{i,m}\},d_{i,j}=D_{i,m}} s_{i,j}\right) \leq \left(\frac{1}{n_{i,m}-2}\sum_{j\in[2N]\setminus\{i,p_{i,m},q_{i,m}\},d_{i,j}=D_{i,m}} \exp(s_{i,j})\right)^{n_{i,m}-2}, \quad (15)$$

thus

$$-\log \frac{\exp\left(\sum\limits_{j\in[2N]\setminus\{i,p_{i,m},q_{i,m}\},d_{i,j}=D_{i,m}} s_{i,j}\right)}{\left(\sum\limits_{k\in[2N]\setminus\{i\},\,d_{i,k}=D_{i,m}} \exp(s_{i,k})\right)^{n_{i,m}-2}} \geq (n_{i,m}-2)\log(1+\exp(\eta_{i,m}) + (n_{i,m}-2)\theta_{i,m}) - (n_{i,m}-2)\log(\theta_{i,m}) \quad (16)$$

By plugging Eqn. (13), Eqn. (14) and Eqn. (16) into Eqn. (12), we have

$$\mathcal{L}_{\text{RNC}} > \frac{1}{2N(2N-1)}\sum_{i=1}^{2N}\sum_{m=1}^{M_i}\left(n_{i,m}\log(1+\exp(\eta_{i,m}) + (n_{i,m}-2)\theta_{i,m}) - \eta_{i,m} - (n_{i,m}-2)\log(\theta_{i,m})\right). \quad (17)$$

Let $h(\theta) := n_{i,m}\log(1+\exp(\eta_{i,m})+(n_{i,m}-2)\theta) - \eta_{i,m} - (n_{i,m}-2)\log(\theta)$. From derivative analysis we know $h(\theta)$ decreases monotonically when $\theta \in \left[1, \frac{1+\exp(\eta_{i,m})}{2}\right]$ and increases monotonically when $\theta \in \left[\frac{1+\exp(\eta_{i,m})}{2}, \exp(\eta_{i,m})\right]$, thus

$$h(\theta) \geq h\left(\frac{1+\exp(\eta_{i,m})}{2}\right) = n_{i,m}\log n_{i,m} + 2\log\frac{1+\exp(\eta_{i,m})}{2} - \eta_{i,m}. \quad (18)$$

By plugging Eqn. (18) into Eqn. (17), we have

$$\mathcal{L}_{\text{RNC}} > \frac{1}{2N(2N-1)}\sum_{i=1}^{2N}\sum_{m=1}^{M_i}\left(n_{i,m}\log n_{i,m} + 2\log\frac{1+\exp(\eta_{i,m})}{2} - \eta_{i,m}\right)$$
$$= L^\star + \frac{1}{2N(2N-1)}\sum_{i=1}^{2N}\sum_{m=1}^{M_i}\left(2\log\frac{1+\exp(\eta_{i,m})}{2} - \eta_{i,m}\right). \quad (19)$$

Then, since $\eta_{i,m} \geq 0$, we have $2\log\frac{1+\exp(\eta_{i,m})}{2} - \eta_{i,m} \geq 0$. Thus, $\forall i \in [2N]$, $m \in [M_i]$,

$$\mathcal{L}_{\text{RNC}} > L^\star + \frac{1}{2N(2N-1)}\left(2\log\frac{1+\exp(\eta_{i,m})}{2} - \eta_{i,m}\right). \quad (20)$$

If $\mathcal{L}_{\text{RNC}} < L^\star + \epsilon \leq L^\star + \frac{1}{2N(2N-1)}\left(2\log\frac{1+\exp(\delta)}{2} - \delta\right)$, then

$$2\log\frac{1+\exp(\eta_{i,m})}{2} - \eta_{i,m} < 2\log\frac{1+\exp(\delta)}{2} - \delta. \quad (21)$$

Since $y(x) = 2\log\frac{1+\exp(x)}{2} - x$ increases monotonically when $x > 0$, we have $\eta_{i,m} < \delta$. Hence, $\forall i \in [2N]$, $j,k \in [2N]\setminus\{i\}$, if $d_{i,j} = d_{i,k} = D_{i,m}$, $|s_{i,j} - s_{i,k}| \leq \eta_{i,m} < \delta$.

Next, we show $s_{i,j} > s_{i,k} + \delta$ if $d_{i,j} < d_{i,k}$ when $\mathcal{L}_{\text{RNC}} < L^\star + \epsilon$.

From Eqn. (4) we have

$$\mathcal{L}_{\text{RNC}} = -\frac{1}{2N(2N-1)}\sum_{i=1}^{2N}\sum_{m=1}^{M_i}\sum_{j\in[2N]\setminus\{i\},\,d_{i,j}=D_{i,m}}\log\frac{\exp(s_{i,j})}{\sum\limits_{k\in[2N]\setminus\{i\},\,d_{i,k}=D_{i,m}}\exp(s_{i,k})}$$
$$+ \frac{1}{2N(2N-1)}\sum_{i=1}^{2N}\sum_{m=1}^{M_i}\sum_{j\in[2N]\setminus\{i\},\,d_{i,j}=D_{i,m}}\log\left(1 + \frac{\sum\limits_{k\in[2N]\setminus\{i\},\,d_{i,k}>D_{i,m}}\exp(s_{i,k}-s_{i,j})}{\sum\limits_{k\in[2N]\setminus\{i\},\,d_{i,k}=D_{i,m}}\exp(s_{i,k}-s_{i,j})}\right), \quad (22)$$

and combining it with Eqn. (5) we have

$$\mathcal{L}_{\text{RNC}} \geq L^\star + \frac{1}{2N(2N-1)} \sum_{i=1}^{2N} \sum_{m=1}^{M_i} \sum_{j \in [2N] \setminus \{i\}, \, d_{i,j} = D_{i,m}} \log \left( 1 + \frac{\sum\limits_{k \in [2N] \setminus \{i\}, \, d_{i,k} > D_{i,m}} \exp(s_{i,k} - s_{i,j})}{\sum\limits_{k \in [2N] \setminus \{i\}, \, d_{i,k} = D_{i,m}} \exp(s_{i,k} - s_{i,j})} \right)$$

$$> L^\star + \frac{1}{2N(2N-1)} \log \left( 1 + \frac{\exp(s_{i,k} - s_{i,j})}{\sum\limits_{l \in [2N] \setminus \{i\}, \, d_{i,l} = d_{i,j}} \exp(s_{i,l} - s_{i,j})} \right),$$

$$(23)$$

$\forall i \in [2N], \, j \in [2N] \setminus \{i\}, k \in \{k \in [2N] \setminus \{i\} \mid d_{i,j} < d_{i,k}\}.$

When $\mathcal{L}_{\text{RNC}} < L^\star + \epsilon$, we already have $|s_{i,l} - s_{i,j}| < \delta, \forall d_{i,l} = d_{i,j}$, which derives $s_{i,l} - s_{i,j} < \delta$ and thus $\exp(s_{i,l} - s_{i,j}) < \exp(\delta)$. By putting this into Eqn. (22), we have $\forall i \in [2N], \, j \in [2N] \setminus \{i\}, k \in \{k \in [2N] \setminus \{i\} \mid d_{i,j} < d_{i,k}\}$,

$$\mathcal{L}_{\text{RNC}} > L^\star + \frac{1}{2N(2N-1)} \log \left( 1 + \frac{\exp(s_{i,k} - s_{i,j})}{n_{i,r_{i,j}} \exp(\delta)} \right), \tag{24}$$

where $r_{i,j} \in [M_i]$ is the index such that $D_{i,r_{i,j}} = d_{i,j}$.

Further, given $\mathcal{L}_{\text{RNC}} < L^\star + \epsilon < L^\star + \frac{1}{2N(2N-1)} \log \left( 1 + \frac{1}{n_{i,r_{i,j}} \exp(\delta + \frac{1}{\delta})} \right)$, we have

$$\log \left( 1 + \frac{\exp(s_{i,k} - s_{i,j})}{n_{i,r_{i,j}} \exp(\delta)} \right) < \log \left( 1 + \frac{1}{n_{i,r_{i,j}} \exp(\delta + \frac{1}{\delta})} \right) \tag{25}$$

which derives $s_{i,j} > s_{i,k} + \frac{1}{\delta}, \forall i \in [2N], \, j \in [2N] \setminus \{i\}, k \in \{k \in [2N] \setminus \{i\} \mid d_{i,j} < d_{i,k}\}.$

Finally, $\forall i \in [2N], \, j, k \in [2N] \setminus \{i\}, s_{i,j} < s_{i,k} - \frac{1}{\delta}$ if $d_{i,j} > d_{i,k}$ directly follows from $s_{i,j} > s_{i,k} + \frac{1}{\delta}$ if $d_{i,j} < d_{i,k}$.

## B  Additional Theoretical Analysis

In this section, we present an analysis based on Rademacher Complexity [39] to substantiate that $\delta$-ordered feature embedding results in a better generalization bound.

A regression learning task can be formulated as finding a hypothesis $h$ to predict the target $y$ from the input $x$. Suppose there are $m$ data points in the training set $\mathcal{S} = \{(x_k, y_k)\}_{k=1}^{m}$. Let $\mathcal{H}_1$ be the class of all possible functions from the input space to the target space.

If a $\delta$-ordered feature embedding is guaranteed with an encoder $f$ mapping $x_k$ to $v_k$, the set of candidate hypotheses can be reduced to all "$\delta$-monotonic" functions $h(x) = g(f(x))$, which satisfy $d(g(v_i), g(v_j)) < d(g(v_i), g(v_k))$ for $s_{i,j} > s_{i,k} + \frac{1}{\delta}$, $d(g(v_i), g(v_j)) = d(g(v_i), g(v_k))$ for $|s_{i,j} - s_{i,k}| < \delta$, and $d(g(v_i), g(v_j)) > d(g(v_i), g(v_k))$ for $s_{i,j} < s_{i,k} - \frac{1}{\delta}$ for any $i, j$ and $k$. We denote the class of all "$\delta$-monotonic" functions by $\mathcal{H}_2$. Note that both $\mathcal{H}_1$ and $\mathcal{H}_2$ contain the optimal hypothesis $h^*$, which satisfies $\forall x, y, h^*(x) = y$.

Further, for a hypothesis set $\mathcal{H}_i$, let $\mathcal{A}_i = \{(l((x_1, y_1); h), ..., l((x_m, y_m); h)) : h \in \mathcal{H}_i\}$ be the loss set for each hypothesis in $\mathcal{H}_i$ with respect to the training set $\mathcal{S}$, where $l$ is the loss function. Let $c_i$ be the upper bound of $|l((x, y); h))|$ for all $x, y$ and $h \in \mathcal{H}_i$. We introduce the Rademacher Complexity [39] of $\mathcal{A}_i$, denoted as $R(\mathcal{A}_i)$. Then, the generalization bound based on Rademacher Complexity says that with a high probability (at least $1 - \epsilon$), the gap between the empirical risk (i.e., training error) and the expected risk (i.e., test error) is upper bounded by $2R(\mathcal{A}_i) + 4c_i \sqrt{\frac{2 \ln(4/\epsilon)}{m}}$.

Since $\mathcal{H}_2 \subset \mathcal{H}_1$, we have $\mathcal{A}_2 \subset \mathcal{A}_1$ and $c_2 \leq c_1$, and from the monotonicity of Rademacher Complexity we have $R(\mathcal{A}_2) \leq R(\mathcal{A}_1)$. Hence, with a $\delta$-ordered feature embedding, the upper bound on the gap between the training error and the test error will be reduced, which leads to better regression performance.

Table 7: **Visualizations of original and augmented data samples on all datasets.**

| Dataset | Original | Augmented | | | |
|---------|----------|-----------|---|---|---|
| AgeDB |  |  |  |  |  |
| TUAB |  |  |  |  |  |
| MPIIFaceGaze |  |  |  |  |  |
| SkyFinder |  |  |  |  |  |

## C  Dataset Details

Five real-world datasets are used in the experiments:

- AgeDB [32, 44] is a dataset for predicting age from face images. It contains 16,488 in-the-wild images of celebrities and the corresponding age labels. The age range is between 0 and 101. It is split into a 12,208-image training set, a 2140-image validation set, and a 2140-image test set.

- TUAB [34, 11] is a dataset for brain-age estimation from EEG resting-state signals. The dataset comes from EEG exams at the Temple University Hospital in Philadelphia. Following Engemann et al. [11], we use only the non-pathological subjects, so that we may consider their chronological age as their brain-age label. The dataset includes 1,385 21-channel EEG signals sampled at 200Hz from individuals whose age ranges from 0 to 95. It is split into a 1,246-subject training set and a 139-subject test set.

- MPIIFaceGaze [51, 52] is a dataset for gaze direction estimation from face images. It contains 213,659 face images collected from 15 participants during their natural everyday laptop use. We subsample and split it into a 33,000-image training set, a 6,000-image validation set, and a 6,000-image test set with no overlapping participants. The gaze direction is described as a 2-dimensional vector with the pitch angle in the first dimension and the yaw angle in the second dimension. The range of the pitch angle is -40° to 10° and the range of the yaw angle is -45° to 45°.

- SkyFinder [31, 7] is a dataset for temperature prediction from outdoor webcam images. It contains 35,417 images captured by 44 cameras around 11am on each day under a wide range of weather and illumination conditions. The temperature range is $-20\,°\mathrm{C}$ to $49\,°\mathrm{C}$. It is split into a 28,373-image training set, a 3,522-image validation set, and a 3,522-image test set.

- IMDB-WIKI [36, 44] is a large dataset for predicting age from face images, which contains 523,051 celebrity images and the corresponding age labels. The age range is between 0 and 186 (some images are mislabeled). We use this dataset to test our method's resilience to reduced training data, performance on transfer learning, and ability to generalize to unseen targets. We subsample the dataset to create a variable size training set, and keep the validation set and test set unchanged with 11,022 images in each.

# D  Details of Data Augmentation

Table 7 shows examples of original and augmented data samples on each dataset. The data augmentations used on each dataset are listed below:

- For `AgeDB` and `SkyFinder`, random crop and resize (with random horizontal flip), color distortions are used as data augmentation;

- For `TUAB`, random crop is used as data augmentation;

- For `MPIIFaceGaze`, random crop and resize (without random horizontal flip), color distortions are used as data augmentation.

# E  Details of Competing Methods

## E.1  End-to-End Regression Methods

We compared with seven end-to-end regression methods:

- $L_1$, MSE and HUBER have the model directly predict the target value and train the model with an error-based loss function, where $L_1$ uses the mean absolute error, MSE uses the mean squared error and HUBER uses an MSE term when the error is below a threshold and an $L_1$ term otherwise.

- DEX [36] and DLDL-V2 [14] divide the regression range of each label dimension into several bins and learn the probability distribution over the bins. DEX [36] optimizes a cross-entropy loss between the predicted distribution and the one-hot ground-truth labels, while DLDL-V2 [14] jointly optimizes a KL loss between the predicted distribution and a normal distribution centered at the ground-truth value, as well as an $L_1$ loss between the expectation of the predicted distribution and the ground-truth value. During inference, they output the expectation of the predicted distribution for each label dimension.

- OR [33] and CORN [40] design multiple ordered thresholds for each label dimension, and learn a binary classifier for each threshold. OR [33] optimizes a binary cross-entropy loss for each binary classifier to learn whether the target value is larger than each threshold, while CORN [40] learns whether the target value is larger than each threshold conditioning on it is larger than the previous threshold. During inference, they aggregate all binary classification results to produce the final results.

For the classification-based baselines, the regression range is divided into small bins, and each bin is considered as a class. Details for each dataset are as follows:

- For `AgeDB`, the age range is $0 \sim 101$ and the bin size is set to 1;

- For `TUAB`, the brain-age range is $0 \sim 95$ and the bin size is set to 1;

- For `MPIIFaceGaze`, the target range is -40 $\sim$ 10 (°) for the pitch angle and -45 $\sim$ 45 (°) for the yaw angle, and the bin size is set to 0.5 for the pitch angle and is set to 1 for the yaw angle;

- For `SkyFinder`, the temperature range is -20 $\sim$ 49 (°C) and the bin size is set to 1.

## E.2  State-of-the-art Representation Learning Methods

We compared with three state-of-the-art representation learning methods:

- SIMCLR [4] is a contrastive learning method that learns representations by aligning positive pairs and repulsing negative pairs. Positive pairs are defined as different augmented views from the same data input, while negative pairs are defined as augmented views from different data inputs.

- DINO [3] is a self-supervised representation learning method using self-distillation. It passes two different augmented views from the same data input to both the student and the teacher networks and maximizes the similarity between the output features of the student network and those of the teacher network. The gradients are propagated only through the student network and the teacher parameters are updated with an exponential moving average of the student parameters.

- SUPCON [25] extends SIMCLR [4] to the fully-supervised setting, where positive pairs are defined as augmented data samples from the same class and negative pairs are defined as augmented data samples from different classes. To adapt SupCon to the regression task, we follow the standard routine of classification-based regression methods to divide the regression range into small bins and regard each bin as a class.

## E.3 State-of-the-art Regression Learning Methods

We also compared with five state-of-the-art regression learning methods:

- LDS+FDS [44] is the state-of-the-art method on the `AgeDB` dataset. It addresses the data imbalance issue by performing distribution smoothing for both labels and features.

- L2CS-NET [1] is the state-of-the-art method on the `MPIIFaceGaze` dataset. It regresses each gaze angle separately and applies both a cross-entropy loss and an MSE loss on the predictions.

- LDE [7] is the state-of-the-art method on the `SkyFinder` dataset. It converts temperature estimation to a classification task, and the class label is encoded by a Gaussian distribution centered at the ground-truth label.

- RANKSIM [17] is a state-of-the-art regression method that proposes a regularization loss to match the sorted list of a given sample's neighbors in the feature space with the sorted list of its neighbors in the label space.

- ORDINAL ENTROPY [50] is a state-of-the-art regression method that proposes a regularization loss to encourage higher-entropy feature spaces while maintaining ordinal relationships.

## F  Details of Experiment Settings

All experiments are trained using 8 NVIDIA TITAN RTX GPUs. We use the SGD optimizer and cosine learning rate annealing [29] for training. The batch size is set to 256. For one-stage methods and encoder training of two-stage methods, we select the best learning rates and weight decays for each dataset by grid search, with a grid of learning rates from $\{0.01, 0.05, 0.1, 0.2, 0.5, 1.0\}$ and weight decays from $\{10^{-6}, 10^{-5}, 10^{-4}, 10^{-3}\}$. For the predictor training of two-stage methods, we adopt the same search setting as above except for adding no weight decay to the search choices of weight decays. For temperature parameter $\tau$, we search from $\{0.1, 0.2, 0.5, 1.0, 2.0, 5.0\}$ and select the best, which is 2.0. We train all one-stage methods and the encoder of two-stage methods for 400 epochs, and the linear regressor of two-stage methods for 100 epochs.

## G  Additional Experiments and Analyses

### G.1 Impact of Model Architectures

In the main paper, we use ResNet-18 as the default encoder backbone for three visual datasets (`AgeDB`, `MPIIFaceGaze` and `SkyFinder`). In this section, we study the impact of backbone architectures on the experiment results. As Table 8 reports, the results of using ResNet-50 as the encoder backbone are consistent with the results using ResNet-18 in Table 2, indicating that our method is compatible with different model architectures.

Table 8: **Evaluation results using ResNet-50 as the encoder backbone for visual datasets**. The results are consistent with the results using ResNet-18 as the encoder backbone.

| | AgeDB | | MPIIFaceGaze | | SkyFinder | |
|---|---|---|---|---|---|---|
| Metrics | MAE$^\downarrow$ | R$^{2\uparrow}$ | Angular$^\downarrow$ | R$^{2\uparrow}$ | MAE$^\downarrow$ | R$^{2\uparrow}$ |
| L1 | 6.49 | 0.830 | 5.74 | 0.748 | 2.88 | 0.863 |
| RNC(L1) | **6.10** | **0.851** | **5.16** | **0.819** | **2.78** | **0.877** |
| | (+0.39) | (+0.021) | (+0.58) | (+0.071) | (+0.10) | (+0.014) |

## G.2 Standard Deviations of Results

In this section, we study the standard deviations of the best results on each dataset with 5 different random seeds. Table 9 shows their average prediction errors and standard deviations. These results are aligned with the results we reported in the main paper.

Table 9: **Average prediction errors and standard deviations of the best results on each dataset.**

| AgeDB: RNC($L_1$) | TUAB: RNC(DLDL-v2) | MPIIFaceGaze: RNC(OR) | SkyFinder: RNC(DLDL-v2) |
|---|---|---|---|
| $6.19 \pm 0.08$ | $7.00 \pm 0.10$ | $5.24 \pm 0.13$ | $2.87 \pm 0.04$ |

## G.3 Impact of Data Augmentation

In this section, we study the impact of data augmentation on RNC. The following ablations are considered:

- RNC($L_1$)*(without augmentation)*: Data augmentation is removed and $\mathcal{L}_{\text{RNC}}$ is computed over $N$ feature embeddings.

- RNC($L_1$)*(one-view augmentation)*: Data augmentation is only performed once for each data point and $\mathcal{L}_{\text{RNC}}$ is computed over $N$ feature embeddings.

- RNC($L_1$)*(two-view augmentation)*: Data augmentation is performed twice to create a two-view batch and $\mathcal{L}_{\text{RNC}}$ is computed over $2N$ feature embeddings.

- $L_1$*(without augmentation)*: Data augmentation is removed and $L_1$ is computed over $N$ samples.

- $L_1$*(with augmentation)*: Data augmentation is performed for each datapoint and $L_1$ is computed over $N$ samples.

Table 10 reports the results on each dataset, which demonstrate that removing data augmentation will result in a decrease in performance for both $L_1$ and RNC($L_1$), and our method outperforms the baseline with or without data augmentation.

Here we further discuss the different roles played by data augmentation in (unsupervised) contrastive learning methods, end-to-end regression learning methods, and our method respectively:

- For (unsupervised) contrastive learning methods, data augmentation is essential for creating the pretext task, which is to distinguish whether the augmented views belong to the same identity or not. Therefore, removing the data augmentation will result in a complete collapse of the model.

- For end-to-end regression learning methods, data augmentation helps the model generalize better and become more robust to unseen variations. However, without augmentation, the model can still perform reasonably well. The improvement brought by the augmentation is often correlated to how much the data augmentation can compensate for the gap between training data and testing data.

- For RNC, data augmentation is also not necessary since our loss contrasts samples according to label distance rather than the identity. Thus, creating augmented views is not crucial to our method. The role of data augmentation for our method is similar to its role in regression learning methods, namely, improving the generalization ability. The experiment results also confirm this: removing data augmentation will lead to a similar drop in performance for both RNC($L_1$) and $L_1$.

Table 10: **Impact of data augmentation.** Data augmentation is not essential for RNC's superior performance.

| Method | Augmentation | AgeDB | TUAB | MPIIFaceGaze | SkyFinder |
|---|---|---|---|---|---|
| $L_1$ | × | 9.53 | 10.24 | 6.93 | 3.14 |
| RNC($L_1$) | × | **8.96** | **9.88** | **6.31** | **2.97** |
| $L_1$ | ✓ | 6.63 | 7.46 | 5.97 | 2.95 |
| RNC($L_1$) | One-view | 6.40 | 7.29 | 5.46 | 2.89 |
| RNC($L_1$) | Two-view | **6.14** | **6.97** | **5.27** | **2.86** |

## G.4 Two-Stage Training Scheme for End-to-End Regression Methods

RNC employs a two-stage training scheme, where the encoder is trained with $\mathcal{L}_{\text{RNC}}$ in the first stage and a predictor is trained using a regression loss on top of the encoder in the second stage. In Table 11, we also train a predictor using $L_1$ loss on top of the encoder learned by the competing end-to-end regression methods and compare with the performance of RNC($L_1$) on the AgeDB dataset. The results show that the two-stage training scheme does not increase the performance of end-to-end regression methods. In fact, it can even be *detrimental* to performance. This is because these methods are not designed for representation learning, and their loss functions are calculated w.r.t. the final model predictions. As a result, there is no guarantee for the representation learned by these methods. This finding validates that the benefit of RNC is due to the proposed $\mathcal{L}_{\text{RNC}}$ and not the training scheme.

Table 11: **Two-stage training results for end-to-end regression methods on AgeDB.** The two-stage training scheme does not increase or even harms the performance of end-to-end regression methods.

| Method | End-to-End | Two-Stage |
|---|---|---|
| $L_1$ | 6.63 | 6.68 |
| MSE | 6.57 | 6.57 |
| HUBER | 6.54 | 6.63 |
| DEX [36] | 7.29 | 7.42 |
| DLDL-v2 [14] | 6.60 | 7.28 |
| OR [33] | 6.40 | 6.72 |
| CORN [40] | 6.72 | 6.94 |
| RNC($L_1$) | − | **6.14** |

## G.5 Training Efficiency

We compute the average wall-clock running time (in seconds) per training epoch on 8 NVIDIA TITAN RTX GPUs for RNC and compare it with SUPCON [25] on all four datasets, as shown in Table 12. Results indicate that the training efficiency of RNC is comparable to SUPCON.

Table 12: **Average wall-clock running time (in seconds) per training epoch for RNC and SUPCON [25] on all datasets.** The training efficiency of RNC is comparable to SUPCON.

| Method | AgeDB | TUAB | MPIIFaceGaze | SkyFinder |
|---|---|---|---|---|
| SUPCON [25] | 23.1 | 25.3 | 69.1 | 55.6 |
| RNC | 26.2 | 27.3 | 75.4 | 61.8 |
| RATIO | 1.13 | 1.08 | 1.09 | 1.11 |

