# OpenReview forum: "Rank-N-Contrast: Learning Continuous Representations for Regression"
_NeurIPS.cc/2023/Conference — NeurIPS 2023 spotlight_

### Official Review · Reviewer_2Kp1 · 2023-06-29

**Soundness:** 2 fair
**Presentation:** 2 fair
**Contribution:** 2 fair
**Rating:** 6
**Confidence:** 2

**Summary:**

The paper proposes the framework Rank-N-Contrast in order to learn regression-aware feature representations. Authors claim that this representation learning mechanism captures the continuous nature of sample orders and helps achieve better performance in downstream regression task.

**Strengths:**

1. The paper provides both experimental results and theoretical proof to support their method.
2. The paper includes relatively comprehensive ablation study and analysis section.

**Weaknesses:**

1. The paper is in lack high-level intuition and in-depth analysis of why the method work and only have low-level interpretation on the experimental results. Following are some example questions I hope authors could have addressed (and go one-step further than just stating their claims): Why preserving order in feature representation help to learn continuous nature? In fig. 3, why discernible pattern is better? What are the theorems trying to convey (my understanding is that feature embedding follows order inherited from the labels)?

2. Although authors have conducted experiments on 4 different datasets (where is the results of IMDB-WIKI b.t.w.?) and with 7 different regression losses, the datasets mostly have scalar responses (except for MPIFaceGaze) and seem repetitive to me. I think the experimental results will be more convincing if authors include experiments with higher-dimensional responses.


**Questions:**

See Weakness section.

**Limitations:**

I don't think authors have adequately addressed their limitations and weaknesses in the paper.

---

> ### Author Rebuttal · Authors · 2023-08-10
>
> Dear Reviewer 2Kp1,
>
> Thank you for your valuable questions and thoughtful feedback. Your comments have helped us to further improve the quality of our paper. However, we believe that there are several **important misunderstandings** which we would like to clarify and address point-by-point. We hope you could re-evaluate the paper based on our clarifications, and would really appreciate it if you could consider updating the review, and raising your score accordingly.
>
> > *Why preserving order in feature representation help to learn continuous nature?*
>
> The continuous nature in data means that the target value is continuous, and thus the target distances between different data samples have **orders**. This intuition motivates us to learn a representation that preserves the order in the label space.
>
> In our **Global Response**, we provide a theoretical analysis based on Rademacher Complexity to demonstrate that a feature embedding that preserves the order in the label space enables a better **generalization bound**, leading to better regression performance. Further, in our experiments, the superior performance (Sec. 5.1), as well as the improved data efficiency, robustness to spurious targets and data corruptions, and generalization to distribution shifts (Sec. 5.2) confirm that preserving order in the feature space *indeed* helps to learn the continuous nature of regression tasks.
>
> To summarize, we’ve justified both **theoretically** and **empirically** that preserving order in feature representation helps to learn continuous targets.
>
> > *In fig. 3, why discernible pattern is better?*
>
> At a high level, Fig.3 illustrates the **correlation** between feature similarities and label similarities for both RnC and L1.  Each entry (i,j) in the matrix refers to the feature similarity between samples i and j.  Samples in the rows and columns of the matrix are ordered according to increased label values, i.e., the first row/column refers to the sample with the smallest label and the last row/column to the sample with the largest label. Thus, the samples that are most similar in the label space are the ones along the diagonal and the least similar in the label space are the two corners farthest away from the diagonal, with other points between the diagonal and these two corners gradually decreasing in their similarity in the label space.  The **more** the feature similarity pattern (i.e., the color pattern in the figure) follows the label similarity pattern (described above), the **higher** the correlation between the similarity in the feature space and the label space. We will make them clearer in the revised paper.
>
> > *What are the theorems trying to convey?*
>
> The theoretical analysis is trying to prove that optimizing the proposed RnC loss will lead to an ordered feature embedding from a theoretical perspective. Thus, we first formalize the description of ordered feature embedding as $\delta$-ordered (Definition 1), and then demonstrate that when the proposed RnC loss is optimized to be sufficiently low, the feature embedding will be $\delta$-ordered (Theorem 3). We also prove that the proposed RnC loss is able to be optimized to be low enough (Theorem 2). Besides, as highlighted in the global response, we further prove that learning $\delta$-ordered feature embeddings can indeed lead to better regression performance based on Rademacher Complexity. Combining all above, we conclude that optimizing the RnC loss will lead to better regression performance.
>
> > *Although ..., the datasets mostly have scalar responses (except for MPIFaceGaze) and seem repetitive to me. I think ... experiments with higher-dimensional responses.*
>
> We want to clarify that the datasets we included in the experiments are **not** repetitive. To comprehensively evaluate the performance over regression tasks, we should not only take into consideration the output dimension, but also other important factors, such as application domain and input dimensions/modalities. The datasets in our evaluation are carefully selected considering all these factors together. They cover:
>
> - Diversity in **application domains**: computer vision (AgeDB), human-computer interaction (MPIIFaceGaze), healthcare (TUAB) and weather monitoring (SkyFinder).
> - Diversity in **input dimensions/modalities**: 2D images (AgeDB, MPIIFaceGaze, SkyFinder), and 1D time series (TUAB).
> - Diversity in **output dimensions**: scalar values (AgeDB, TUAB, SkyFinder) and higher-dimensional vectors (MPIIFaceGaze).
>
> Please note that the level of diversity in our datasets is significantly high. Nonetheless, if the reviewer has specific suggestions on datasets with higher output dimensions, we are happy to include more results.
>
> > *Where is the results of IMDB-WIKI?*
>
> We apologize for the confusion. As we mentioned in Appendix B, we used IMDB-WIKI only for the **analysis**: testing our method’s resilience to reduced training data, performance on transfer learning, and the ability to generalize to unseen targets. We didn’t include it in the main results because we already incorporated AgeDB in the main results for the task of age estimation from face images; in addition, the age labels in AgeDB have been manually cleaned by other researchers while the age labels in IMDB-WIKI contain noise [1]. We will make this point clearer in the revised paper and properly refer to them in the main text.
>
> [1] Moschoglou et al. AgeDB: the first manually collected, in-the-wild age database. CVPR Workshop 2017.
>
> ---
> We hope the above clarifications have addressed all of your concerns, and made you more confident about the novelty, significance, and completeness of our paper. If you have more questions or suggestions, please do not hesitate to discuss with us. We thank you again for your time and feedback. We hope you could re-evaluate the paper based on our clarifications, and would really appreciate it if you could consider updating the review, and raising your score accordingly.

---

> > ### Comment · Reviewer_2Kp1 · 2023-08-10
> > **Raise score to 6**
> >
> > Thanks for the authors making clarifications and further explaining the details of the experiments. After carefully reading the authors' rebuttal and other reviewers' comments, I am generally satisfied with the responses to my concerns. I decide to raise my score to 6.

---

> > > ### Author Response · Authors · 2023-08-12
> > > **Thank you very much for your feedback**
> > >
> > > Thank you very much for your feedback. We are glad to learn that our response has addressed your concerns and that you decided to raise your score to 6. We noticed however that the score and review haven’t yet been updated. Thus, we would like to kindly request that you update your score and review in the system to reflect your decision to raise the score to 6.
> > >
> > > Once again, thank you very much for your time and effort, and please do not hesitate to let us know if you have any further questions or comments about the paper.

---

### Official Review · Reviewer_dBkT · 2023-07-04

**Soundness:** 4 excellent
**Presentation:** 4 excellent
**Contribution:** 3 good
**Rating:** 8
**Confidence:** 4

**Summary:**

The paper introduces a deep learning method, Rank-N-Contrast, for regression tasks. This method aims to capture continuity in data, something existing methods struggle with. The authors define a concept of $\delta$-ordered feature embedding and show theoretically that if Rank-N-Contrast loss is minimized, feature embeddings will be $\delta$-ordered. The proposed method excels in several regression tasks.

**Strengths:**

This paper provides a novel perspective to regression tasks using deep learning models, a well-studied and widely-acknowledged problem setting. The paper's unique approach lies in its application of contrastive learning methods, which is a departure from conventional techniques that alter the loss function or incorporate specific regularization.

The paper sets itself apart by offering an extensive comparison to existing methodologies, using both theoretical and empirical results from regression and representation learning tasks. This comprehensive approach underlines the proposed method's potential and relevance.

The ubiquity of regression tasks in the realm of deep learning models implies that this paper's approach could contribute meaningfully to the field. Furthermore, the paper is well-composed, making it easy for readers to grasp the fundamental concept of the study.

**Weaknesses:**

The paper positions "regression-aware representation" as vital, but a slight disparity between the theoretical and experimental results is apparent.

Theoretically, it suggests that a deep learning model can fulfill "regression-aware embedding" characteristics by minimizing the RNC loss appropriately. Experimentally, the paper exhibits consistent performance enhancements over existing approaches in both regression tasks and representation learning frameworks. Furthermore, it confirms that the implementation of RNC provides resilience against data corruption, improves performance with limited training data, and boosts transfer learning.

While the paper implies the potential to secure $\delta$-ordered feature embeddings by reducing RNC loss, and evidences steady performance improvement in real-world data contexts, the precise relationship between the continual characteristics of the "regression-aware representations" and the observed performance improvements is not fully clear yet. Additional discussion around this subject could offer more insights and, thereby, enrich our comprehension.

**Questions:**

1. **About Embedding Dimensions:** The learned representations in Fig. 1 are impressive, but the embedding dimension isn't specified. Were these embeddings trained directly in two dimensions, or were they first embedded in a high-dimensional space and then visualized with methods like UMAP? If UMAP was used, are the apparent clustering and lack of continuity in L1 and SupCon embeddings an artifact of the visualization process, or do these characteristics persist in the high-dimensional embeddings?
2. **About the Construction of RNC loss:** In l.104, just before Equation 1, it's stated that the normalized likelihood of $v_j$ "can be written as …" Is Equation 1 derived from some definition or axiom? At least the normalized likelihood is “defined” in the reference [41].
3. **About Trainability:** Theorem 3 claims that if RNC loss is sufficiently minimized, a desirable representation can be obtained. But how much can RNC loss be practically minimized? Can any insights be provided from a theoretical standpoint on the trainability of the proposed loss?

**Limitations:**

One potential limitation is the difference in the number of training epochs between the one-stage and two-stage methods. According to Appendix l.629, the one-stage methods were trained for 400 epochs, whereas the two-stage methods, including the proposed approach, were trained for a total of 500 epochs. This longer training period for the proposed method could potentially be contributing to the improvements observed in table 2. A further analysis or an ablation study might help to confirm whether these enhancements are genuinely due to the method itself and not merely a consequence of the extended training period.

---

> ### Author Rebuttal · Authors · 2023-08-10
>
> Dear Reviewer dBkT,
>
> Thank you very much for acknowledging the novelty and the contributions of our work. We sincerely appreciate the time and effort you have dedicated to evaluating our work. In the following, we address your concerns in detail.
>
> > *The precise relationship between the continual characteristics of the "regression-aware representations" and the observed performance improvements is not fully clear yet. Additional discussion around this subject could offer more insights and, thereby, enrich our comprehension.*
>
> Thanks for the valuable suggestion. Indeed, theoretical connections **do** exist between regression-aware representations and the observed enhancement in performance: In our **Global Response**, we provide a theoretical analysis based on Rademacher Complexity to show that ***$\delta$-ordered feature embedding leads to better generalization bound***. We will expand on the analysis therein and include it in the revised manuscript as a new theorem following Theorem 3.
>
> > ***About Embedding Dimensions**: The learned representations in Fig. 1 are impressive, but the embedding dimension isn't specified. Were these embeddings trained directly in two dimensions, or were they first embedded in a high-dimensional space and then visualized with methods like UMAP? If UMAP was used, are the apparent clustering and lack of continuity in L1 and SupCon embeddings an artifact of the visualization process, or do these characteristics persist in the high-dimensional embeddings?*
>
> Thanks for pointing this out, and we apologize for the confusion. The embeddings in Fig.1 are first embedded in a 512-dimensional embedding space, then visualized using UMAP. The clustering and lack of continuity are **unlikely** to be artifacts, because if the points are clustered / far apart in the UMAP visualization, it suggests that those points were close to / distant from each other in the high-dimensional space as well [29].
>
> Furthermore, we calculated the Spearman’s rank correlation coefficient and the Kendall rank correlation coefficient between label similarities and feature similarities on that dataset in **Sec. 3 - Feature Ordinality**, where the feature similarities are computed from the **original** 512-dimensional feature vectors. The results in Table 1 confirm that the feature similarities learned by our method have significantly higher correlations with the label similarities than those by the L1 loss, which further verifies that the embeddings learned by RnC are indeed more continuous. We will make these points clearer in the revised paper.
>
> > ***About the Construction of RNC loss**: In l.104, just before Equation 1, it's stated that the normalized likelihood of v_j  "can be written as …" Is Equation 1 derived from some definition or axiom? At least the normalized likelihood is “defined” in the reference [41].*
>
> Thank you for the great question. As stated in l.99, the **form** of the normalized likelihood is commonly used by related metric learning literatures [16, 43], where the likelihood is modeled to increase exponentially with respect to the feature similarity. Furthermore, we are inspired by [41] – where the denominator contains a **subset** of samples for ranking purposes – to introduce an adaptive set $\mathcal{S_{i, j}}$ that contains the samples of higher rank than $v_j$ given $v_i$, and define a **customized** likelihood $\mathbb{P}(v_j | v_i, \mathcal{S_{i, j}})$ accordingly that is suitable for our problem settings.
>
> We apologize for the potential confusion and will make these points clearer in the revised paper.
>
> > ***About Trainability**: Theorem 3 claims that if RNC loss is sufficiently minimized, a desirable representation can be obtained. But how much can RNC loss be practically minimized? Can any insights be provided from a theoretical standpoint on the trainability of the proposed loss?*
>
> Thanks for the insightful comment. Actually, Theorem 2 proves that the RnC loss can be *arbitrarily* close to its lower bound, which means that the RnC loss can be sufficiently minimized for any $0 < \delta < 1$ from a **theoretical** perspective.
>
> However, in practice, for any loss function, how much the loss can be minimized **empirically** depends a lot on the model, task, and data, such as whether the model capacity is large enough, whether the input contains sufficient information about the task, and whether the label is clean or not. Thus it usually cannot be *simply and/or universally* guaranteed. We will make a remark in the revised paper to discuss this point.
>
> > *One potential limitation is the difference in the number of training epochs between the one-stage and two-stage methods .... A further analysis or an ablation study might help to confirm whether these enhancements are genuinely due to the method itself and not merely a consequence of the extended training period.*
>
> Thanks for pointing out the difference between training epochs. In fact, we have already included that ablation study in **Appendix F.4 of the submission**. In this section, we adopted the two-stage training scheme for each of the one-stage methods, i.e., training the predictor for 100 more epochs on top of the encoder which was trained for 400 epochs. The results show that the two-stage training scheme does **not** help improve the performance of those one-stage methods, which further validates that the benefit of RnC stems from the proposed loss function rather than the training scheme / number of training epochs.
>
> ---
> We thank you again for your time and feedback. We hope that our response has adequately answered your questions, and would lead to a favorable increase of the score. We are happy to discuss more if you have any further questions.

---

> > ### Comment · Reviewer_dBkT · 2023-08-22
> > **Response**
> >
> > Thank you for giving us comprehensive responses to all the questions. Roughly all my questions were addressed by the authors' feedback. I would leave one point of slight concern.
> >
> > Regarding the use of UMAP visualization, the authors stated as follows
> >
> > > The clustering and lack of continuity are unlikely to be artifacts, because if the points are clustered / far apart in the UMAP visualization, it suggests that those points were close to / distant from each other in the high-dimensional space as well [29].
> >
> > I would like to leave a different point of view on this comment. Indeed, dimensionality reduction methods such as UMAP are designed to preserve distance relationships in the original space as much as possible. In practical situations, however, UMAP visualization can often reveal superficially spurious cluster structures, even in artificial data that is inherently random and structureless. Since such a phenomenon depends on the number and dimension of the data and the hyperparameters of the UMAP, it would be good to have a simple ablation study that would eliminate such a possibility.
> >
> > Overall, my assessment of this paper remains the same. This paper is well-written, and its value is clear. I believe it is well worthy of being accepted, while one minor point I mentioned above still remains. Therefore, I would like to keep my initial score.

---

> > > ### Author Response · Authors · 2023-08-22
> > > **Further response to the UMAP question**
> > >
> > > Thank you for your feedback and for your opinion that the paper is worthy of being accepted. As suggested by the reviewer, we will include an ablation study of the number / dimension of the data and the hyperparameters of the UMAP in the revised paper. Here, due to the limited time, we provide some preliminary results that show the structure is **not** due to artifacts:
> > >
> > > - **Ablation of the number of data samples**: In Fig. 6 of the main paper, we show the UMAP visualization using a 10-webcam subset of SkyFinder dataset, whereas Fig. 1 uses the full dataset, which contains 44 webcams. The structure of the visualization in Fig. 6 is consistent with Fig. 1, indicating that the difference in their number of data samples did not eliminate the structure.
> > >
> > > - **Ablation of the dimension of feature embeddings**: In the PDF attached in the Global Response, we generated the same plots for L1 and RnC as in Fig. 1, but using ViT-Small as the backbone, whose feature dimension is 384. The structure of the visualization in the PDF is consistent with Fig. 1, despite their difference in the number of dimensions (384 vs. 512).
> > >
> > > - **Ablation of the hyperparameters of UMAP**: Following your suggestions, we explored the impact of major UMAP hyperparameters, such as the number of neighbors (`n_neighbors`) and the minimum distance between embedded points (`min_dist`). Specifically, we tried `n_neighbors` from {5, 10, 20, 50, 100} and `min_dist` from {0, 0.25, 0.5, 0.8, 0.99} (In Fig. 1 we use the default parameters in UMAP, where `n_neighbors = 15` and `min_dist = 0.1`). Unfortunately, we are not able to include the figure at this stage, however, we would like to let you know the structures of visualizations are still consistent with Fig. 1 among all of these hyperparameters: L1 and SupCon embeddings are fragmented while RnC embeddings are continuous.
> > >
> > > Once again, we thank the reviewer for the constructive feedback and insightful suggestions. We hope the above results will address your concerns and help you be more confident about our paper. We will stress this point and provide a comprehensive ablation study in the revised paper.

---

### Official Review · Reviewer_wKSy · 2023-07-04

**Soundness:** 3 good
**Presentation:** 3 good
**Contribution:** 3 good
**Rating:** 7
**Confidence:** 4

**Summary:**

The authors present a new loss for representation learning in regression, RNC (Rank-n-contrast).  RNC can be seen as the SupCon loss adapted to the regression setting, where the labels given are not hard class labels but rather continuous regression labels.  In SupCon the negatives for each example are members of other classes and the positives come from members of the same class.  In RNC positive pairs are formed with every example to the anchor, and the negatives are those examples with larger label distances to the anchor than the positives.  By doing this they are able to train a representation for the data that is continuous in nature, which is hypothesized to represent the data better.

**Strengths:**

1.   I believe the method is sufficiently original as a nontrivial adaptation of the SupCon methodology to the regression setting.  The designation of what the positives and negatives are for the RNC contrastive loss are well motivated and I can see why the loss would give the kind of continuous representations that it does.

2.  The method seems to be pretty simple to implement.  I particularly like that the RNC loss is the only loss the authors pretrained with and it wasn't some highly tuned composition of many different losses.  It gives me more confidence that the RNC loss is providing the gain in performance.

3.  The theory is simple and well-motivated in that it proves the loss is doing what the authors are claiming it does.  The theory combined with the visualization of the representations give me confidence that the representations are in fact ordered continuously.

4.  The experimental evaluation is fairly thorough.  The standard questions are answered, i.e. standard deviation error bars, how does it perform against other pretraining tasks, how does a plain two-stage training compare, whether augmentation is important, etc.

**Weaknesses:**

1.  I think the experimental evaluation can be improved to improve the generality of the method.  Currently the datasets that are evaluated are on computer vision regression tasks, which confines the evaluation to just ResNet-based models when we compare against the different losses and training methods in the paper.  In particular, one area where regression tasks are popular is the tabular setting.  I would be very interested to see a comparison between RNC and tree-based methods on popular tabular methods.  For examples of models and datasets to compare on the authors can check https://arxiv.org/abs/2106.11959 (not my paper btw)

2.  It is great to see that the method is able to consistently perform well when compared against other baselines.  But as someone who is not familiar with these particular regression tasks I am not sure whether the magnitude of improvement is substantial.  Could the authors help me understand the scale of improvement?

3.  Returning to my point on how the evaluation is restricted to ResNet-based architectures, I wonder if given that ResNet was designed to classify images that it has some inductive bias on clustering sets of images based on semantic similarities, which is what causes the disjointed representations in Figure 1 (left).  Expanding past the computer vision domain would be very valuable in making the contribution more impactful and general.

**Questions:**

Questions and possible directions of improvement are listed under "Weaknesses".

**Limitations:**

I believe this was sufficient.

---

> ### Author Rebuttal · Authors · 2023-08-10
>
> Dear Reviewer wKSy,
>
> Thanks for the constructive comments and insightful feedback. We are glad that you found the method novel and simple to implement, the theory simple and well-motivated and the evaluation thorough. Here we address your concerns one by one.
>
> > *Currently the datasets that are evaluated are on computer vision … I would be very interested to see a comparison between RNC and tree-based methods on popular tabular methods.*
>
> First, we would like to clarify that **not** all datasets included in our evaluation are image datasets - TUAB is a dataset consisting of EEG signals (i.e., time-series data). Besides, our method is **not** restricted to ResNet, but rather applicable to various deep architectures, including Transformer-based architectures. We chose ResNet as our main backbone because ResNet is the most commonly used architecture for mainstream regression methods and tasks [1, 2, 3, 4]. Here we further ran experiments with **ViT-Small** as the backbone on two datasets. The results (metric: MAE) are shown in the table below.
>
> ||AgeDB|SkyFinder|
> |-|:-:|:-:|
> |L1|10.18|3.63|
> |**RnC(L1)**|**9.58**|**3.50**|
>
> Note that the performance with ViT could be worse than that with ResNet since ViT typically requires larger amounts of training data and usually performs worse on smaller datasets [5]. Nevertheless, RnC(L1) still performs *significantly better* than L1, showing the generality of RnC.
>
> Besides, following the reviewer’s suggestion, we further conducted experiments on a **tabular** dataset and compared RnC with tree-based methods and competitive deep models. Specifically, we followed the same evaluation protocols in [6] and evaluated on a subset (10% random subsampled due to time and computation limit) of Microsoft (MI, search queries) dataset [7]. The results are shown in the table below.
>
> |Method|RMSE|
> |-|:-:|
> |CatBoost|0.758 $\pm$ 4.8e-4|
> |XGBoost|0.756 $\pm$ 6.7e-4|
> |ResNet|0.762 $\pm$ 5.3e-4|
> |**RnC(ResNet)**|0.756 $\pm$ 6.4e-4|
> |FT-Transformer|0.760 $\pm$ 8.9e-4|
> |**RnC(FT-Transformer)**|**0.753 $\pm$ 7.7e-4**|
>
> The results show that applying RnC to the deep models *significantly improved* their performance and allowed them to *outperform* the popular tree-based methods on the tabular dataset. We will add the above results and discussions in the revised paper.
>
> > *As someone who is not familiar with these particular regression tasks I am not sure whether the magnitude of improvement is substantial. Could the authors help me understand the scale of improvement?*
>
> Thanks for the great question. Here we discuss the improvement scale for each dataset:
>
> - **AgeDB**: The incorporation of RnC reduces the prediction error by 5.8% on average for all regression methods in Table 2, and Table 3 shows that our performance gain against the best SOTA method is 0.31 years, while the performance gap between the best and the second-best SOTA method is 0.02 years.
>
> - **TUAB**: The incorporation of RnC reduces the prediction error by 9.3% on average for all regression methods in Table 2, and Table 3 shows that our performance gain against the best SOTA method is 0.31 years, while the performance gap between the best and the second-best SOTA method is 0.05 years.
>
> - **MPIIFaceGaze**: The incorporation of RnC reduces the prediction error by 11.7% on average for all regression methods in Table 2, and Table 3 shows that our performance gain against the best SOTA method is 0.18 degrees, while the performance gap between the best and the second-best SOTA method is 0.05 degrees.
>
> - **SkyFinder**: The incorporation of RnC reduces the prediction error by 7.0% on average for all regression methods in Table 2, and Table 3 shows that our performance gain against the best SOTA method is 0.06 degrees Celsius, while the performance gap between the best and the second-best SOTA method is 0.01 degree Celsius.
>
> We believe that the performance gain from the proposed RnC framework is *significant* and *substantial*, and we will expand upon these explanations in the revision.
>
> > *I wonder if given that ResNet was designed to classify images that it has some inductive bias on clustering sets of images based on semantic similarities, which is what causes the disjointed representations in Figure 1 (left).*
>
> First, we would like to clarify that the three plots in Fig.1 are generated with the **same** architecture and only differ by the **losses** used to train them, which indicates that it is the different losses that lead to different structures (continuous or fragmented) in the representations. Second, regarding the reviewer’s concern on whether it is the inductive bias in ResNet that leads to the fragmented representations in Fig.1, we generated the same plots for L1 and RnC using **ViT-Small** as the backbone (**Please see the PDF attached in the Global Response**), which reveals similar structures and verifies that the presence of fragmented representations in existing general regression learning schemes stems from their inability to capture the underlying continuous order between samples.
>
> [1] Yang et al. Delving into deep imbalanced regression. ICML 2021.
>
> [2] Gong et al. Ranksim: Ranking similarity regularization for deep imbalanced regression. ICML 2022.
>
> [3] Zhang et al. Improving deep regression with ordinal entropy. ICLR 2023.
>
> [4] Engemann et al. A reusable benchmark for brain-age prediction from M/EEG resting-state signals. NeuroImage 2022.
>
> [5] Zhu et al. Understanding Why ViT Trains Badly on Small Datasets: An Intuitive Perspective. Arxiv 2023.
>
> [6] Gorishniy et al. Revisiting Deep Learning Models for Tabular Data. NeurIPS 2021.
>
> [7] Qin et al. Introducing LETOR 4.0 datasets. Arxiv 2013.
>
> ---
> We hope our response has thoroughly addressed your concerns, and would really appreciate it if you could consider raising your score accordingly. If you have any further questions or suggestions, please do not hesitate to share them. We are eager to engage in further discussions with you.

---

> > ### Comment · Reviewer_wKSy · 2023-08-11
> > **The rebuttal has addressed my concerns.**
> >
> > I have raised my score accordingly.

---

### Official Review · Reviewer_D7S9 · 2023-07-06

**Soundness:** 3 good
**Presentation:** 4 excellent
**Contribution:** 4 excellent
**Rating:** 8
**Confidence:** 3

**Summary:**

The authors discuss the benefits of contrastive learning for learning structured representations in a regression setting. While contrastive losses are typically formulated in terms of “similar” and “dissimilar” examples, the authors make use of the extra information conveyed by the continuous target label. They show how adding the RnC contrastive term to training is beneficial for a suite of high-dimensional regression tasks.


**Strengths:**

This is a technically strong paper that, in my opinion, makes a contribution towards the important and understudied problem of representation learning for regression. The qualitative results [Fig 1] are compelling. The proposed method is intuitive and the paper is well written. The theoretical analysis and extensive empirical results suggest to me that the proposed method will be of interest to NeurIPS attendees. I was especially interested in the suggestion that contrastive training yields more robust representations [lines 259--279], although I think further work will be needed in the future to validate this finding in more sophisticated OOD generalization settings.


**Weaknesses:**

To my knowledge the technical contributions here are sound. However, after reading the paper, I found myself a bit concerned about how neural net-based regression models are going to be used in the near future. And I think the paper would benefit from a head-on discussion of the potential ethical issues at play.

Regression is notoriously difficult for neural networks (for example, many baseline methods convert the regression into a classification problem). A technological advance here could open up entirely new application areas. However, the use of datasets involving human faces suggest to me that some of these new applications could bring up new ethical concerns as well. While Predicting age and gaze direction from face images (as done in this paper) might seem reasonable, there are existing critiques in the literature [e.g. https://ir.lawnet.fordham.edu/cgi/viewcontent.cgi?article=1804&context=iplj, https://medium.com/@blaisea/physiognomys-new-clothes-f2d4b59fdd6a, https://dl.acm.org/doi/pdf/10.1145/3375627.3375820] arguing that predicting other target variables, such as a continuous proxy for emotion, are ethically fraught. In the authors’ opinion, are there application areas where RnC should *not* be applied?

I think the paper would benefit from a discussion of these issues. While the lack of such discussion (the broader impact section is, frankly, rather boilerplate) represents a weakness in my view, it probably should not stand in the way of the paper’s acceptance, since it is more of a critique of the whole subfield rather than this specific paper. However, I am going to request an ethics review in order to get a second opinion about this.

The related works are generally well covered. One exception is the recent C-mixup paper [https://arxiv.org/abs/2210.05775], which also uses label similarity in regression to group “similar” data points. However their approach uses these similarities for data augmentation rather than contrastive learning. A discussion of how the two approaches differ would benefit the reader.


**Questions:**

* How is the temperature parameter \tau tuned? [line 106]


**Limitations:**

Limitations and broader impacts are discussed [Sec. 5.4, App. H]. However, given that images of faces are used frequently in the experiments, I think that a more complete discussion about broader impacts, especially as it relates to task definition for regression, should be included (see “Weaknesses” above and my request for an ethics review below).

---

> ### Author Rebuttal · Authors · 2023-08-10
>
> Dear Reviewer D7S9,
>
> Thank you very much for your valuable feedback. We are delighted to see that you found the method intuitive, the results compelling and the paper well-written, and we wish to express our gratitude for bringing the ethics considerations to our attention. Here we address your concerns one by one.
>
> > *Given that images of faces are used frequently in the experiments, I think that a more complete discussion about broader impacts, especially as it relates to task definition for regression, should be included.*
>
> Thank you for the dedicated discussion about ethical considerations. We certainly agree that these are essential points that deserve being addressed in the main body of the paper. We have **revised our Broader Impact and Limitations sections**, as shown below, and will move them to the **main body** of the revised manuscript. We hope that the revised sections address your concerns.
>
> **Broader Impacts.**
> *We introduce a novel framework designed to enhance the performance of generic deep regression learning. We believe this will significantly benefit regression tasks across various real-world applications. Nonetheless, several potential risks warrant discussion. First, when the framework is employed to regress sensitive personal attributes such as intellectual capabilities, health status, or financial standing from human data (like facial images or physiological signals), there's a danger it might reinforce or even introduce new forms of bias. Utilizing the method in these contexts could inadvertently justify discrimination or the negative targeting of specific groups. Second, in contrast to handcrafted features which are grounded in physical interpretations, the feature representations our framework learns can be opaque. This makes it difficult to understand and rationalize the model’s predictions, particularly when trying to determine if any biases exist. Third, when our method is trained on datasets that do not have a balanced representation of minority groups, there's no assurance of its performance on these groups being reliable. It is essential to recognize that these ethical concerns apply to deep regression (and classification) models at large, not solely our method. However, the continuous nature of our representation which facilitates interpolation and extrapolation might inadvertently make it more tempting to justify such unethical applications. Anyone seeking to implement or use our proposed method should be mindful of these concerns. Both our specific method and deep regression models, in general, should be used cautiously to avoid situations where their deployment might contribute to unethical outcomes or interpretations.*
>
> **Limitations.**
> *Our proposed method presents some limitations. Firstly, the technique cannot discern spurious or incidental correlations between the input and the target within the dataset. As outlined in the Broader Impact section, this could result in incorrect conclusions potentially promoting discrimination or unjust treatment when utilized to deduce personal attributes. Future research should delve deeper into the ethical dimensions of this issue and explore strategies to ensure ethical regression learning. A second limitation is that our evaluation primarily focuses on general regression accuracy metrics (e.g., MAE) without considering potential disparities when evaluating specific subgroups (e.g., minority groups). Given that a regression model's performance can vary across demographic segments, subgroup analysis is an avenue that warrants exploration in subsequent studies. Lastly, our approach learns continuous representations by contrasting samples against one another based on their ranking in the target space, necessitating label information. To adapt it for representation learning with unlabeled data, our framework will need some modifications, which we reserve for future work.*
>
> We certainly welcome any further suggestions from the reviewer, and are more than happy to incorporate them to make the statements more comprehensive.
>
> > *The recent C-mixup paper also uses label similarity in regression to group “similar” data points. A discussion of how the two approaches differ would benefit the reader.*
>
> Thanks for pointing out the missing reference. We sincerely apologize for the oversight during the submission phase. We will cite and discuss the C-mixup paper in our revised manuscript. Here is a draft of the discussion to be added to the **Related Work** section:
>
> *C-mixup [1] leverages label similarity for regression tasks. Specifically, it adapts the original mixup [2] data augmentation technique for regression learning by adjusting the sampling probability of the mixed pairs according to the label similarities. In contrast, our method contrasts samples against each other based on the rankings of label similarities. It is also worth noting that our method is orthogonal and complementary to data augmentation algorithms for regression, such as C-mixup.*
>
> > *How is the temperature parameter \tau tuned? [line 106]*
>
> As discussed in Appendix E, we performed standard hyper-parameter search for the temperature parameter $\tau$ in {0.1, 0.2, 0.5, 1.0, 2.0, 5.0} and selected one with the best performance, which is 2.0.
>
> [1] Yao et al. C-mixup: Improving generalization in regression. NeurIPS 2022.
>
> [2] Zhang et al. Mixup: Beyond Empirical Risk Minimization. ICLR 2018.
>
> ---
> We hope our response has addressed all of your concerns and can lead to a favorable increase of the score. Please feel free to let us know if you have other questions or suggestions. We are more than willing to discuss more with you.

---

> > ### Comment · Reviewer_D7S9 · 2023-08-10
> > **author rebuttal**
> >
> > I read the rebuttal and the other reviews. For now I will keep my score the same, which reflects my belief that the paper would be a very nice addition to the conference. If an ethics review is added later I will read and consider its contents.

---

### Official Review · Reviewer_6JYz · 2023-07-13

**Soundness:** 4 excellent
**Presentation:** 4 excellent
**Contribution:** 4 excellent
**Rating:** 7
**Confidence:** 4

**Summary:**

In this paper, the authors proposed a novel framework that learns continuous representations for regression problems, by contrasting samples against each other based on the rankings induced by the target values. The proposed method is evaluated on several regression tasks and the results show that the proposed method can achieve competitive performance compared to the state-of-the-art methods.

**Strengths:**

1. This paper tackles a very interesting problem of representation fragmentation in deep regression models. The proposed method is simple and very effective to learn continuous representations that fit the regression task. It is very interesting to see that the traditional methods learn representations clustered based on spurious targets (i.e., camera location in the SkyFinder dataset) while the proposed method learns a nice continuous representation that captures the target values.

2. The authors present a theoretical analysis of the proposed loss function and show that it can learn delta-ordered feature embeddings when sufficiently trained.

3. The proposed loss function is conceptually simple and easy to implement. The proposed method is thoroughly evaluated on several regression tasks and the results show that the proposed method can achieve competitive performance compared to the state-of-the-art methods. The authors further conducted ablation studies to show the effectiveness of the proposed method under data corruption, unseen or spurious targets.

**Weaknesses:**

1. While it is nice to see the proposed method leads to delta-ordered feature embeddings. It would be nice if the authors can further theoretically connect the delta-ordered feature embeddings to the final performance of the regression task. This would help to justify that the delta-ordered feature embeddings are indeed useful for the regression task.

2. For the remark for Thm 3, to achieve delta-ordered features for the entire dataset, is it that we need to optimize all batches to achieve low enough loss? Is there any guarantee/insight that there is a feasible solution that can achieve low enough loss for all batches?

3. It seems the IMDB-WIKI performance is missing in Table 2.

**Questions:**

See the comments in "weaknesses".

**Limitations:**

Yes.

---

> ### Author Rebuttal · Authors · 2023-08-10
>
> Dear Reviewer 6JYz,
>
> Thanks for your constructive comments and insightful questions. We are delighted to see that you appreciate the contributions of our work. Below, we address your concerns in detail.
> > *It would be nice if the authors can further theoretically connect the delta-ordered feature embeddings to the final performance of the regression task.*
>
> Thank you for the valuable suggestion. The delta-ordered feature embeddings can *indeed* be theoretically connected to the final performance: In our **Global Response**, we provide an analysis based on Rademacher Complexity to show that ***$\delta$-ordered feature embedding leads to better generalization bound***. We will expand on the theoretical analysis therein and include it as a new theorem following Theorem 3 in the revised paper.
>
> > *For the remark for Thm 3, to achieve delta-ordered features for the entire dataset, is it that we need to optimize all batches to achieve low enough loss? Is there any guarantee/insight that there is a feasible solution that can achieve low enough loss for all batches?*
>
> Thanks for the great question. We do not need to conduct optimizations for all batches, which is also practically impossible. In fact, one should consider the training process as a cohesive whole, which is effectively optimizing the **expectation** of the loss over all possible random batches. In addition, Markov's inequality [1] guarantees that when the expectation of the loss is optimized to be sufficiently low, the loss on any batch will be low enough with a high probability. We will add this point to the revised paper.
>
> > *It seems the IMDB-WIKI performance is missing in Table 2.*
>
> We apologize for the confusion. As we mentioned in Appendix B, we used IMDB-WIKI only for the **analysis**: testing our method’s resilience to reduced training data, performance on transfer learning, and the ability to generalize to unseen targets. We didn’t include it in the main results because we already incorporated AgeDB in the main results for the task of age estimation from face images; in addition, the age labels in AgeDB have been manually cleaned by other researchers while the age labels in IMDB-WIKI contain noise [2]. We will make this point clearer in the revised paper and properly refer to them in the main text.
>
> [1] Grimmett & Stirzaker. Probability and random processes. Oxford University Press 2020.
>
> [2] Moschoglou et al. AgeDB: the first manually collected, in-the-wild age database. CVPR Workshop 2017.
>
> ---
> We hope that our response has addressed all of your concerns and offered any needed clarifications, and that you may consider a favorable increase of the score. Please do not hesitate to discuss with us if you have other comments. We are always happy to take any questions or suggestions.

---

> > ### Comment · Reviewer_6JYz · 2023-08-20
> > **Response**
> >
> > Thank you for the clarifications. I have read the response and other reviews. I keep my current score unchanged.

---

### Author Rebuttal · Authors · 2023-08-10

We are grateful to all the reviewers for the time and effort they invested in reviewing our paper. It is heartening to note that the reviewers found:
- The paper addresses an **important** (D7S9), **ubiquitous** (dBkT), and **interesting** (6JYz) problem.
- The proposed method is **novel** (wKSy, dBkT), **well-motivated** (wKSy), **intuitive** (D7S9), and **easy to implement** (6JYz, wKSy).
- The proposed method is **justified theoretically** (6JYz, D7S9, wKSy, dBkT, 2Kp1), and the theories are **well-motivated** (wKSy).
- The empirical evaluations are **comprehensive** (6JYz, D7S9, wKSy, dBkT, 2Kp1), and the results are **compelling** (6JYz, D7S9).
- The paper is **well-written** and **easy-to-follow** (D7S9, dBkT).

We made a concerted effort to provide a comprehensive response to each reviewer, with point-to-point answers following each review. We hope that our response adequately addresses the reviewers’ concerns and would be happy to answer any additional questions you may have.

---
In this **Global Response** section, we would like to answer a common question raised by Reviewers 6JYz, dBkT and 2Kp1:
> *How does the delta-ordered feature embeddings relate to final performance gain for the regression tasks from a theoretical perspective?*

We thank the reviewers for highlighting this insightful question, which helps to further enhance the completeness of our paper. Learning an ordered feature embedding can *indeed* boost the performance of the regression task.
Below, we present an analysis based on Rademacher Complexity [1] to substantiate that ***$\delta$-ordered feature embedding results in a better generalization bound***:

Specifically, regression learning can be formulated as finding a hypothesis $h$ to predict the target $y$ from the input $x$. Suppose there are $m$ data points in the training set $\mathcal{S}=\\{(x_k, y_k)\\}^m_{k=1}$. Let $\mathcal{H}_1$ be the class of all possible functions from the input space to the target space.

If a $\delta$-ordered feature embedding is guaranteed with an encoder $f$ mapping $x_k$ to $v_k$, the set of candidate hypotheses can be reduced to all "$\delta$-monotonic" functions $h(x) = g(f(x))$, which satisfy $\forall i, j$ and $k$, $d(g(v_i), g(v_j)) < d(g(v_i), g(v_k))$ for $s_{i,j} > s_{i,k} + \frac{1}{\delta}$, $d(g(v_i), g(v_j)) = d(g(v_i), g(v_k))$ for $|s_{i, j} - s_{i, k}| < {\delta}$, and $d(g(v_i), g(v_j)) > d(g(v_i), g(v_k))$ for $s_{i,j} < s_{i,k} - \frac{1}{\delta}$, where $d(\cdot,\cdot)$ is the target distance measure and $s_{i, j}$ is the feature similarity between $v_i$ and $v_j$. We denote the class of all "$\delta$-monotonic" functions by $\mathcal{H}_2$. Note that the optimal hypothesis, i.e., $\forall x, y, h^*(x) = y$, is in both $\mathcal{H}_1$ and $\mathcal{H}_2$.

Further, for a hypothesis set $\mathcal{H}_i$, let $\mathcal{A}_i = \\{(l((x_1, y_1); h), ..., l((x_m, y_m); h)): h\in\mathcal{H}_i\\}$ be the loss set for each hypothesis in $\mathcal{H}_i$ with respect to the training set $\mathcal{S}$, where $l$ is the loss function. Let $c_i$ be the upper bound of $|l((x, y); h))|$ for all $x, y$ and $h \in \mathcal{H}_i$.
We introduce the Rademacher Complexity [1] of $\mathcal{A}_i$, denoted as $R(\mathcal{A}_i)$.
Then, from the generalization bound based on Rademacher Complexity [1], we have, with a high probability (at least $1-\epsilon$), the gap between the empirical risk (i.e., training error) and the expected risk (i.e., test error) is upper bounded by $2R(\mathcal{A}_i) + 4c_i\sqrt{\frac{2\ln(4/\epsilon)}{m}}$.

Since $\mathcal{H}_2 \subset \mathcal{H}_1$, we have $\mathcal{A}_2 \subset \mathcal{A}_1$ and $c_2 \leq c_1$, and from the monotonicity of Rademacher Complexity we have $R(\mathcal{A}_2) \leq R(\mathcal{A}_1)$. Hence, with a $\delta$-ordered feature embedding, the upper bound on the gap between the training error and the test error will be reduced, which leads to better regression performance.

---
To put it more intuitively, fitting an ordered feature embedding **reduces** the **complexity** of the regressor, which enables **better generalization ability** from training to testing, and ultimately leads to the **final performance gain**.

Relatedly, we note that the enhanced generalization ability is further **empirically verified** in our paper (see **Sec. 5.2** in our main paper). Specifically, if not constrained, the learned feature embeddings could capture spurious or easy-to-learn features that are not generalizable to the real continuous targets (see **Robustness to Spurious Targets**). Such property also leads to better robustness to data corruptions, better resilience to reduced training data, and better generalization to unseen targets.

To summarize, learning an ordered feature embedding can indeed lead to better performance for regression tasks. We will describe these results formally as a new theorem following Theorem 3 in the revised paper. We believe the new results will make our paper more significant and comprehensive.

[1] Shalev-Shwartz & Ben-David. Understanding machine learning: From theory to algorithms. Cambridge University Press 2014.

---
We hope our response has adequately addressed the reviewers’ question(s), and would really appreciate it if the reviewers could consider raising their scores after reading our response.  We are happy to take any further questions from the reviewers.

---

### Decision · Program_Chairs · 2023-09-21

**Decision:**

Accept (spotlight)

**Comment:**

This paper proposes a strategy for learning embeddings for regression tasks. Reviewers appreciated the originality of the method, and the empirical and theoretical results. Although several concerns (including ethical) were raised, they seemed to properly addressed during the review period. All reviewers unanimously suggest that this paper be accepted.